


# Green Edge ice camp campaigns: understanding the processes controlling the under-ice Arctic phytoplankton spring bloom

Philippe Massicotte[1], Rémi Amiraux[1,2], Marie-Pier Amyot[1], Philippe Archambault[1,3], Mathieu Ardyna[4,5], Laurent Arnaud[6], Lise Artigue[7], Cyril Aubry[1], Pierre Ayotte[8,9,10], Guislain Bécu[1], Simon Bélanger[11], Ronald Benner[12], Henry C. Bittig[4,13], Annick Bricaud[4], Éric Brossier[14], Flavienne Bruyant[1], Laurent Chauvaud[2], Debra Christiansen-Stowe[15], Hervé Claustre[4], Véronique Cornet-Barthaux[16], Pierre Coupel[1], Christine Cox[14], Aurelie Delaforge[17], Thibaud Dezutter[1], Céline Dimier[18], Florent Dominé[1], Francis Dufour[1], Christiane Dufresne[19,20,3], Dany Dumont[19,20,3], Jens Ehn[17], Brent Else[21], Joannie Ferland[1], Marie-Hélène Forget[1], Louis Fortier[1], Martí Galí[1], Virginie Galindo[19,20,3], Morgane Gallinari[2], Nicole Garcia[16], Catherine Gérikas-Ribeiro[22,23], Margaux Gourdal[1], Priscilla Gourvil[24], Clemence Goyens[25], Pierre-Luc Grondin[1], Pascal Guillot[26], Caroline Guilmette[1], Marie-Noëlle Houssais[27], Fabien Joux[28], Léo Lacour[1], Thomas Lacour[29], Augustin Lafond[16], José Lagunas[1], Catherine Lalande[1], Julien Laliberté[1], Simon Lambert-Girard[1], Jade Larivière[1], Johann Lavaud[1], Anita LeBaron[1], Karine Leblanc[16], Florence Le Gall[22], Justine Legras[16], Mélanie Lemire[8,30,10], Maurice Levasseur[1,3], Edouard Leymarie[4], Aude Leynaert[2], Adriana Lopes dos Santos[31], Antonio Lourenço[32], David Mah[31], Claudie Marec[1,33], Dominique Marie[34], Nicolas Martin[32], Constance Marty[14], Sabine Marty[35], Guillaume Massé[1], Atsushi Matsuoka[1], Lisa Matthes[17], Brivaela Moriceau[2], Pierre-Emmanuel Muller[14], Christopher-John Mundy[17], Griet Neukermans[1,4], Laurent Oziel[1,4], Christos Panagiotopoulos[16], Jean-Jacques Pangazi[14], Ghislain Picard[36], Marc Picheral[4], France Pinczon du Sel[14], Nicole Pogorzelec[17], Ian Probert[24], Bernard Queguiner[16], Patrick Raimbault[16], Joséphine Ras[4], Eric Rehm[1], Erin Reimer[1], Jean-François Rontani[16], Søren Rysgaard[17], Blanche Saint-Béat[1], Makoto Sampei[37], Julie Sansoulet[1], Sabine Schmidt[38], Richard Sempéré[16], Caroline Sévigny[39], Yuan Shen[40,41], Margot Tragin[34], Jean-Éric Tremblay[1], Daniel Vaulot[34,31], Gauthier Verin[1], Frédéric Vivier[32], Anda Vladoiu[42,43], Jeremy Whitehead[21], and Marcel Babin[1]

[1]UMI Takuvik, CNRS/Université Laval, Québec, QC Canada
[2]Univ Brest, CNRS, IRD, Ifremer, LEMAR, F-29280 Plouzane, France
[3]Québec-Océan
[4]Sorbonne Université, CNRS, Laboratoire d'Océanographie de Villefranche, LOV, F-06230 Villefranche-sur-Mer, France
[5]Department of Earth System Science, Stanford University, Stanford, CA, 94305, USA
[6]UMR 5001, IGE, CNRS, Grenoble, France
[7]LEGOS, University of Toulouse, CNRS, CNES, IRD, UPS, 31400 Toulouse, France
[8]Axe Santé des populations et pratiques optimales en santé, Centre de recherche du CHU de Québec - Université Laval
[9]Centre de toxicologie du Québec, INSPQ
[10]Département de médecine sociale et préventive, Université Laval, Québec QC Canada
[11]Département de Biologie, Chimie et Géographie (groupes BORÉAS et Québec-Océan), Université du Québec à Rimouski, 300 allé des Ursulines, Rimouski, QC, G5L 3A1





[12]University of South Carolina, Department of Biological sciences, Columbia, SC 29208 USA

[13]Leibniz Institute for Baltic Sea Research Warnemünde, IOW, Rostock-Warnemünde, Germany

[14]Independent collaborator

[15]Institut nordique du Québec, Université Laval, Québec, QC Canada

[16]Aix-Marseille Univ., Université de Toulon, CNRS, IRD, MIO, UM110, Marseille, 13288, France

[17]Centre for Earth Observation Science, University of Manitoba, Winnipeg, Manitoba, Canada

[18]FR3761, Institut de la Mer de Villefranche, CNRS, 06230 Villefranche-sur-mer, France

[19]Institut des sciences de la mer de Rimouski

[20]Université du Québec à Rimouski

[21]Department of Geography, University of Calgary, Calgary, Alberta, Canada

[22]CNRS, Sorbonne Université, UMR7144, Team ECOMAP, Station Biologique de Roscoff,Roscoff, France

[23]GEMA Center for Genomics, Ecology & Environment, Universidad Mayor, Camino La Pirámide, 5750, Huechuraba, Santiago, Chile

[24]Sorbonne Université, CNRS, FR2424, Centre de Ressources Biologiques Marines, Station Biologique de Roscoff, France

[25]Royal Belgian Institute of Natural Sciences (RBINS), Operational Directorate Natural Environment, 29 Rue Vautierstraat, 1000 Brussels, Belgium

[26]Québec Océan & Amundsen Science, Université Laval, Québec, QC Canada

[27]LOCEAN, CNRS/Sorbonne Université/IRD/MNHN, 4 place Jussieu, F-75005 Paris, France

[28]Sorbonne Université, CNRS, Laboratoire d'Océanographie Microbienne (LOMIC), Observatoire Océanologique de Banyuls, 66650 Banyuls/mer, France

[29]IFREMER, Physiology and Biotechnology of Algae Laboratory, rue de l'Ile d'Yeu, 44311, Nantes, France

[30]Institut de biologie intégrative et des systèmes

[31]Asian School of the Environment, Nanyang Technological University, 50 Nanyang Avenue, Singapore 639798

[32]LOCEAN-IPSL,CNRS, Sorbonne Université, Paris, France

[33]Univ Brest, CNRS, IUEM, UMS3113, F-29280 Plouzane, France

[34]UMR 7144, Sorbonne Université and CNRS, Station Biologique, 29680 Roscoff, France

[35]Norwegian institute for water research, Gaustadalleen 21, 0349 Oslo, Norway

[36]Institut des Géosciences de l'Environnement 54, rue Molière 38402 - Saint Martin d'Hères, France

[37]Faculty of fisheries sciences, Hokkaido University, Hakodate, Japan

[38]UMR CNRS 5805 EPOC - OASU, Université de Bordeaux, 33615 PESSAC CEDEX, FRANCE

[39]Environnement et changement climatique Canada

[40]School of the Earth, Ocean and Environment, University of South Carolina, Columbia, South Carolina, 29208, USA

[41]Present address: Ocean Sciences Department, University of California, Santa Cruz, California, 95064, USA

[42]LOCEAN-IPSL, Sorbonne Université, Paris, France

[43]Applied Physics Laboratory, University of Washington, Seattle, Washington, USA

**Correspondence:** Philippe Massicotte (philippe.massicotte@takuvik.ulaval.ca)

**Abstract.** The Green Edge initiative was developed to investigate the processes controlling the primary productivity and the fate of organic matter produced during the Arctic phytoplankton spring bloom (PSB) and to determine its role in the ecosystem. Two field campaigns were conducted in 2015 and 2016 at an ice camp located on landfast sea ice southeast of Qikiqtarjuaq Island in Baffin Bay (67.4797N, 63.7895W). During both expeditions, a large suite

of physical, chemical and biological variables was measured beneath a consolidated sea ice cover from the surface to the bottom at 360 m depth to better understand the factors driving the PSB. Key variables such as temperature, salinity, radiance, irradiance, nutrient concentrations, chlorophyll-a concentration, bacteria, phytoplankton





and zooplankton abundance and taxonomy, carbon stocks and fluxes were routinely measured at the ice camp. Here, we present the results of a joint effort to tidy and standardize the collected data sets that will facilitate their

reuse in other Arctic studies. The dataset is available at https://www.seanoe.org/data/00487/59892/ (Massicotte et al., 2019a).

## 1   Introduction

In the Arctic Ocean, the phytoplankton spring bloom (PSB) initiates the period of highest biomass primary production of the year (Sakshaug, 2004; Perrette et al., 2011; Ardyna et al., 2013). Although it was discovered that the PSB

may occur more extensively and more frequently beneath a consolidated ice-pack (Arrigo et al., 2012, 2014; Assmy et al., 2017), only a small number of research initiatives (e.g., Fortier et al., 2002; Galindo et al., 2014; Mundy et al., 2009, 2014; Wassmann et al., 1999; Gosselin et al., 1997) have been investigating the processes controlling the Arctic PSB in the ice-covered water column. Additionally, ice algal communities play an important role within the Arctic food web and for the carbon export to the benthos during the winter-spring transition (Leu et al., 2015). However,

primary production within the Arctic ice-pack is still poorly understood. The Green Edge project was conceived in an effort to better understand the Arctic PSB from the level of fundamental physical, chemical and biological processes to that of their interactions within the ecosystem, and at spatial scales ranging from local to pan-Arctic. Besides studying each major component of the processes controlling Arctic PSB, another objective of Green Edge was to investigate its impact on the nutrient and carbon dynamics within the ecosystem. A total of three Green Edge

campaigns were conducted: two ice camp campaigns on landfast sea ice in 2015 and 2016, and an oceanographic cruise aboard the *CCGS Amundsen* in Baffin Bay in 2016. In this article, we present an overview of an extensive and comprehensive data set acquired during two surveys conducted at the Green Edge ice camp.

## 2   Study area, environmental conditions and sampling strategy

The field campaigns were conducted on landfast sea ice southeast of the Qikiqtarjuaq Island in Baffin Bay (67.4797N,

63.7895W, Fig. 1) in 2015 (April 24 - July 17) and in 2016 (April 20 to July 27). These periods were chosen in order to capture the dynamics of the sea-ice algae and phytoplankton spring blooms, from bloom initiation to termination. The field operations took place at a location (the "ice camp") south of the Qikiqtarjuaq Island where the water depth is 360 m. Continuous records of wind speed and air temperature were made with a meteorological station (Automated Meteo Mat equipped with temperature (HC2S3) and wind (05305-L) sensors (Campbell Scientific) po-

sitioned near (< 100 m) the tent (Polarhaven, Weatherhaven) in which water sampling was carried out. During the sampling periods, the study site experienced changes in snow cover and ice thickness (Fig. 2). In 2015, the snow and ice thickness varied between 2-40 cm (mean = 21 cm) and 103-136 cm (mean = 121) respectively. In 2016, the snow and ice thickness varied between 0.3-49 cm (mean = 19 cm) and 106-149 cm (mean = 128 cm) respectively. For both years, snowmelt began at the beginning of June and lasted for approximately two to three weeks (Oziel





et al., 2019). Water sampling was usually carried out every two days through a 1×1 m hole in the ice pack shielded
by the tent. For the analysis of nutrient concentration, photosynthetic parameters, primary production, chlorophyll
a (chl a), phytoplankton taxonomy and carbon stocks such as dissolved organic carbon (DOC), particulate organic
carbon (POC), water samples were collected at 1.5, 5, 10, 20, 40 and 60 m using 10 or 20-L Niskin bottles. Details
about specific measurements such as zooplankton and bacteria abundances are provided in the following sections.

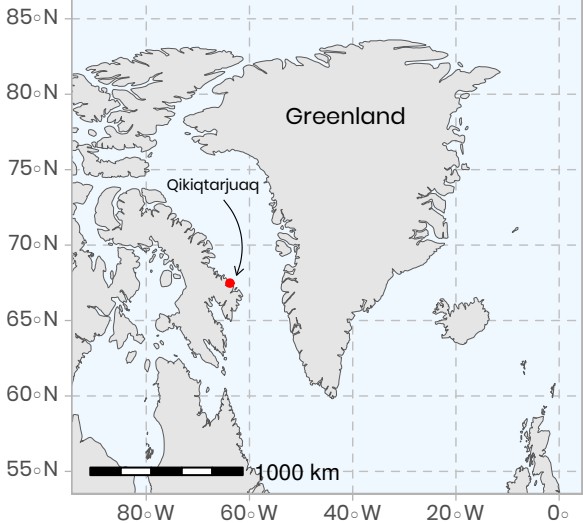

**Figure 1.** Location of the ice camp located near the Qikiqtarjuaq Island in the Baffin Bay. Projection used: EPSG-4326.



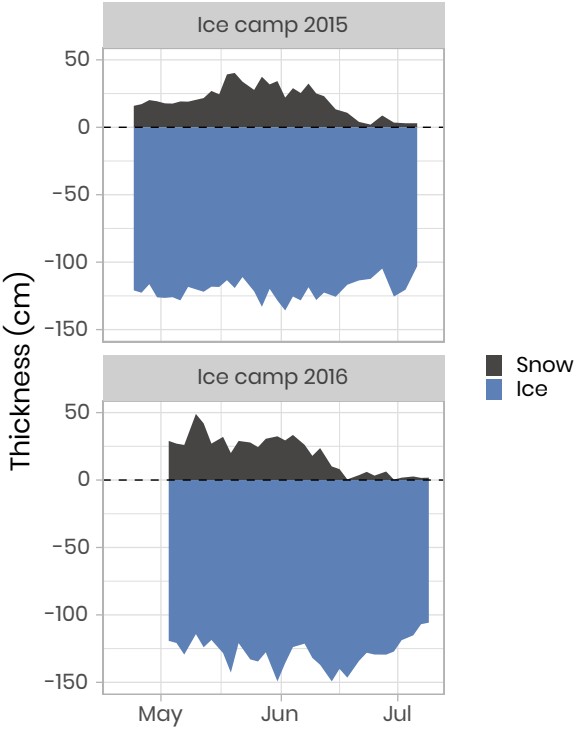

**Figure 2.** Temporal evolution of the snow and sea-ice thickness for both ice camp missions. The dashed horizontal line represents the snow/ice interface.

## 3   Data quality control and data processing

Different quality control procedures were adopted to ensure the integrity of the data. First, the raw data were visually screened to eliminate errors originating from the measurement devices, including sensors (systematic or random) and errors inherent from measurement procedures and methods. Statistical summaries such as average, standard deviation and range were computed to detect and remove anomalous values in the data. Then, data were

50 checked for duplicates and remaining outliers. Once raw measurements were cleaned, data were structured and regrouped into plain text comma-separated (CSV) files. Each of these files was constructed to gather variables of the same nature (ex.: nutrients). In each of these files, a minimum number of variables (columns) were always included so the different data sets can be easily merged together (Table 1). More than 120 different variables have been measured during the Green Edge landfast-ice expeditions. The complete list of variables is presented in Table

2 and detailed metadata information can be found on the LEFE-CYBER online repository http://www.obs-vlfr.fr/proof/php/GREENEDGE/greenedge.php. The processed and tidied version of the data is hosted at SEANOE (SEA scieNtific Open data Edition) under the CC-BY license (https://www.seanoe.org/data/00487/59892/, Massicotte et al. (2019a)). In the following sections, we present a subset of these variables along with the methods used to collect





and measure them. For each of these variables, time series or vertical profiles are used to describe the data. Data
cleaning and visualization were performed with R 3.6.1 (R Core Team, 2019). The code used to produce the figures
and the analysis presented in this paper is available under the GNU GPLv3 licence https://github.com/PMassicotte/
greenedge-icecamp-data-paper. The code used to process and tidy the data provided by each researcher is also
publicly available https://gitlab.com/Takuvik/greenedge-database under the GNU GPLv3 licence.

## 4  Data description: an overview

### 4.1  Physical data

Conductivity, temperature and depth (CTD) vertical profiles were measured using a Sea-Bird SBE19plusV2 CTD sys-
tem (factory calibrated prior to the expedition) deployed from inside the Polarhaven tent between the surface and
a 350 m depth. The data were post-processed according to the standard procedures recommended by the manu-
facturer and averaged into 1-m vertical bins. During the sampling periods, salinity was generally greater than 31.5
70  g kg$^{-1}$ (range: 4–34.4 g kg$^{-1}$). Flushes of freshwater at the ocean surface due to snow/ice melt started slowly at the
beginning of June with the largest peaks/pulses taking place late June when salinity decreased to approximately 4 g
kg$^{-1}$ (Fig. 3).

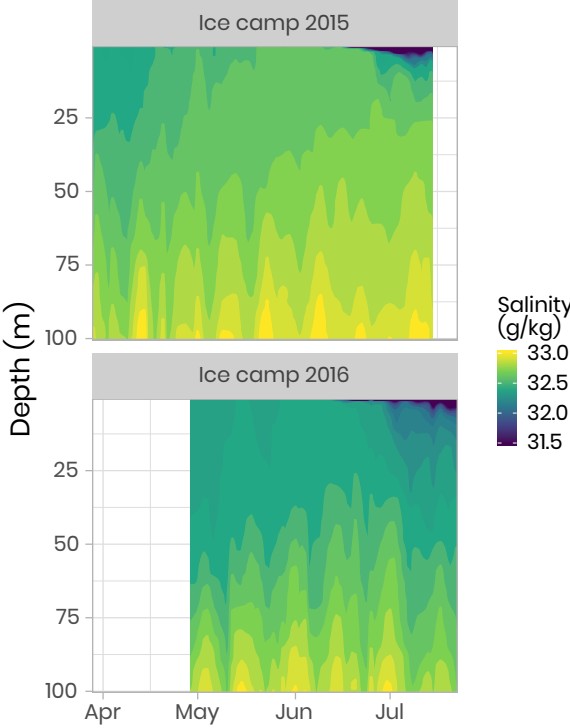

**Figure 3.** Temporal evolution of the salinity in the first 100 meters of the water column for both campaigns. Note that for visualization, salinity below 31.5 g kg$^{-1}$ have been binned to 31.5 g kg$^{-1}$. Note that salinity as low as 4 g kg$^{-1}$ was observed during flushes of freshwater at the ocean surface due to snow/ice melt (dark blue color in the figure).

Ocean current profiles in the water column were measured using a downward-looking 300 kHz Sentinel Workhorse Acoustic Doppler Current Profiler (ADCP, RDI Teledyne) mounted directly beneath the sea ice bottom. The study site was dominated by seawater originating from the Arctic Ocean modulated by spring-neap tidal cycles (14 days) and semidiurnal M2 periods (≈12.4 hours). Vertical profiles of water column turbulence were measured on June 23 of 2016 during a spring tidal cycle (≈12.4) using a self-contained autonomous microprofiler (SCAMP, Precision Measurement Engineering, California, U.S.A.). The turbulence profile (i.e. a median profile of the rate of dissipation of turbulent kinetic energy, $\epsilon$) showed a mixing layer depth of about 20–25 m characterized by an elevated dissipation rate with values above 10$^{-8}$ W kg$^{-1}$. The reader is referred to the paper by Oziel et al. (2019) for detailed methods, visualization and discussion of the CTD, SCAMP and ADCP data.

Vertical profiles (surface to 200 m) of CTD and bio-optical properties were measured every hour during a M2 tidal cycle measured on June 9, 2016 (an example of modelled surface tidal height versus time is shown in supplementary Fig. A1). These observations (Fig. 4) illustrate that internal tidal waves caused large vertical isopycnal displacements (20-30 m) of all observed physical and biogeochemical properties below 50 m depth across the semi-diurnal M2 period. Hence, as vertical profiles of physical and bio-optical variables were measured at approximately the same time





each day, properties (assuming they follow a conservative mixing behaviour) will appear to be vertically displaced. Therefore, when comparing properties from vertical profiles taken at the ice camp, we suggest that comparisons of profile variables should be made on isopycnal (constant density) coordinates, rather than depth coordinates (Fig.

4).

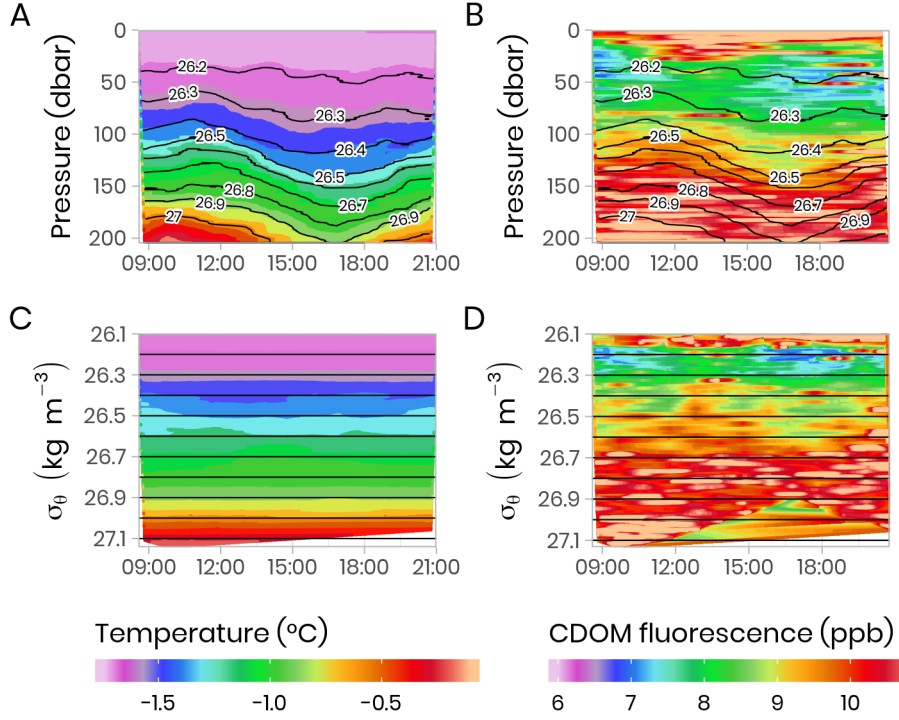

**Figure 4.** Temporal evolution of physical (temperature) and bio-optical (CDOM fluorescence) variables with superimposed lines of potential density anomaly ($\sigma_\theta$, kg m$^{-3}$) during a 13-h tidal cycle. Surface tidal height versus time at Qikiqtarjuaq is shown in blue. (**A-B**) Plotted versus pressure coordinates (equivalent to depth in meters). (**C-D**) The same data plotted versus potential density anomaly $\sigma_\theta$ coordinates (kg m$^{-3}$). The tidal survey was performed on 2015-06-09.

## 4.2 Underwater bio-optical data

### 4.2.1 Radiance and irradiance measurements with ICE-Pro

A total of 173 and 89 vertical light profiles were measured in 2015 and 2016, respectively, using a factory-calibrated ICE-Pro (an ice-floe version of the C-OPS, or Compact-Optical Profiling System, from Biospherical Instruments Inc.).

The ICE-Pro was equipped with radiometers for both downward plane irradiance ($E_d$, W m$^{-2}$ nm$^{-1}$) and either upward irradiance ($E_u$, W m$^{-2}$ nm$^{-1}$) in 2015 or upward radiance ($L_u$, W m$^{-2}$ sr$^{-1}$ nm$^{-1}$) in 2016. The profiles were taken at two sites, separated by approximately 40 m. In order to perform the profiles, the ICE-Pro was deployed through

auger holes that had been drilled at distances of 82 and 113 m from the tent and cleaned of ice chunks. Once the ICE-Pro was underneath the ice layer, fresh clean snow was shovelled back into the hole to avoid, as much as
possible, having a bright spot above the sensors (see supplementary Fig. B1 and Table B1). The frame was then manually lowered at a rate of approximately 0.3 m s$^{-1}$. The above-surface reference sensor was fixed on a steady tripod installed approximately 2 m above the ice surface and above all neighbouring camp features. Data processing and validation were performed using a protocol inspired by that of Smith1984, which is now used by several space agencies. Measurements were taken between 380 and 875 nm at 19 discrete spectral wavebands. Vertical
profiles were usually measured in duplicates or triplicates. Time series of daily photosynthetically active radiation (PAR) at the sea-ice/water interface (1.3 m depth) are shown in Fig. 5. In 2016, PAR started to increase rapidly in the second week of May, compared to early June in 2015. Overall, PAR at 1.3 m in the water column was also greater in 2016 than in 2015 and reached the threshold of 0.415 mol of photons m$^{-2}$ d$^{-1}$, above which light is sufficient for net growth (Letelier et al., 2004), a few days earlier. Further information about in situ underwater irradiance and
radiance measurements can be found in Massicotte et al. (2018).

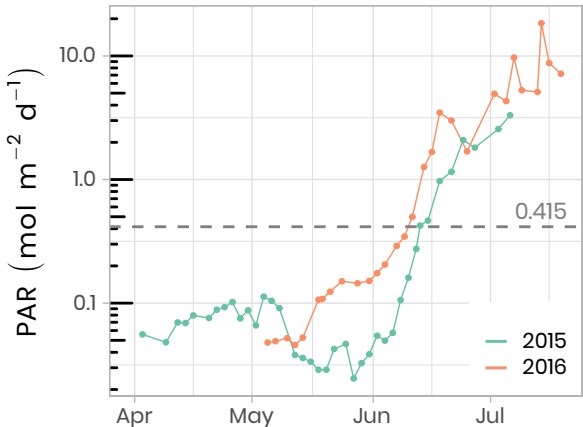

**Figure 5.** Temporal evolution of daily photosynthetically available radiation (PAR) at the sea-ice/water interface (1.3 m depth) for both ice camp missions. The horizontal dashed line shows the 0.415 mol photons m$^{-2}$ d$^{-1}$ threshold often used in the literature as the minimum light requirement for primary production.

### 4.2.2 Underwater photos and videos of ice bottom

Several vertical profiles to 30 m were performed using a GoPro Hero 4 camera mounted on the ICE-Pro and pointing up, towards the ice bottom (see Fig. B1 and Table B1). Still images were captured every five seconds during descent, as well as a video was taken of the complete descent. These photos and videos were used for a qualitative as-
115 sessment of the pronounced spatial and temporal heterogeneity of the under-ice environment and the associated water column nekton community between the two profiling locations.





### 4.2.3 Irradiance measurements with TriOS

To quantify the impact of the heterogeneous radiation field under sea ice on irradiance measurements, replicated spectral irradiance profiles were collected beneath landfast sea ice from 5 May to 8 June 2015 and from 14 June to
120 4 July 2016. The replicates were made on each sampling day, under different surface conditions. In 2015, measurements were performed prior to melt onset, under different snow depths. In 2016, measurements began after the onset of snowmelt and were performed beneath sea ice with a wet snow cover, shallow melt ponds and white ice. The deployed sensor array consisted of a surface reference radiometer, which recorded incident downwelling planar irradiance, $E_d(0,\phi)$, and three radiometers attached to a custom-built double-hinged aluminum pole (under-ice
L-arm) to measure downwelling planar irradiance, $E_d(z,\phi)$, downwelling scalar irradiance, $\mathring{E}_d(z,\phi)$, and upwelling scalar irradiance, $\mathring{E}_u(z,\phi)$. These four hyperspectral radiometers (two planar RAMSES-ACC and two scalars RAMSES-ASC, TriOS GmbH, Germany) measured pressure and tilt internally and recorded irradiance spectra in the wavelength range from 320 to 950 nm at a resolution of 3.3 nm (190 channels). Transmitted irradiance was recorded along with vertical profiles by lowering the L-arm manually through a 20-inches auger hole with a winch and 1.5-
130 m aluminum poles extensions. In 2015, 17 vertical profiles were collected in 0.4 - 0.5-m depth steps from the ice bottom to a water depth of 18 m. In 2016, 11 profiles were recorded to a depth of 20 m under different sea ice surface conditions. Differences between planar and scalar PAR measurements were used to derive the downwelling average cosine, μd, an index of the angular structure of the downwelling under-ice radiation field which, in practice, can be used to convert between downwelling scalar, $\mathring{E}_d$, and planar, $E_d$, irradiance. The average cosine was smaller
prior to snowmelt in 2015 compared to after snowmelt (≈0.6 vs. 0.7), when melt ponds covered the ice surface in 2016 (Fig. 6). Further details about the sampling procedure, data processing and results can be found in Matthes et al. (2019).



**Figure 6.** (**A**) Under-ice vertical profiles of downwelling planar and scalar irradiance at 442 nm, 532 nm and for PAR. Note the log scale for the irradiance measurements. (**B**) Calculated downwelling average cosine (unitless) was measured beneath snow-covered sea ice on 16 May 2015, beneath bare ice on 20 June 2016 and beneath a melt pond on 4 July 2016.



### 4.2.4 Inherent optical properties (IOP)

IOPs measurements were made using an optical frame equipped with the physical and bio-optical sensors that
were factory calibrated before each field campaign. A Seabird SBE-9 CTD measured temperature, salinity, and pressure. A WetLabs AC-S was used for spectral beam attenuation ($c$, m$^{-1}$) and total absorption ($a$, m$^{-1}$) between 405 and 740 nm, and a BB9 (WetLabs) and a BB3 (WetLabs) were utilized for backscattering coefficients ($bb$, m$^{-1}$) between 440 and 870 nm. During both campaigns, pure water calibration was performed for the AC-S sensor on each sampling day and linear regression as a function of time was computed for each wavelength of absorption and attenuation
signals. Then, the offset applied during the data processing was taken on this linear regression at the exact date of the measurement. Figure 7 shows two vertical profiles of attenuation coefficients at different wavelengths acquired during pre-bloom and bloom conditions in 2016. One can see that during the bloom, attenuation increased markedly in the 0-50 m surface layer due to higher phytoplankton biomass.

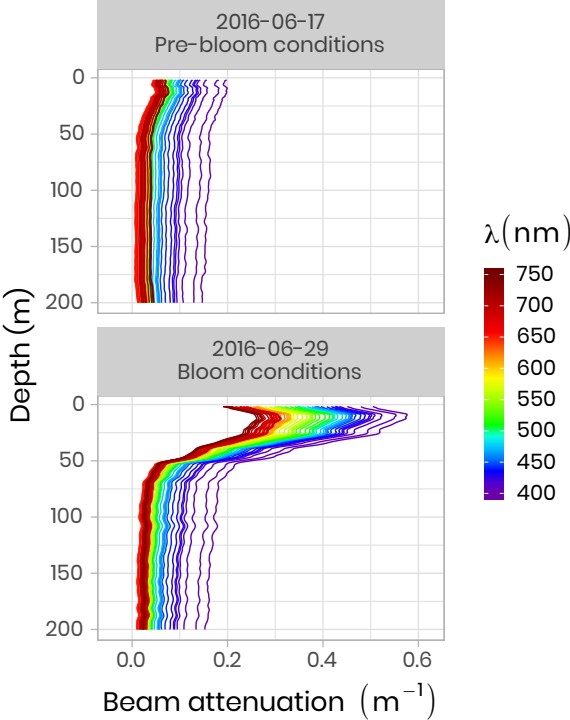

**Figure 7.** Beam attenuation coefficients ($c$, m$^{-1}$) measured in 2016 using an ACS before and during the phytoplankton bloom. Note that the colors of the lines correspond to wavelength frequencies.




### 4.2.5   Other optical measurements

Other optical variables measured during both field campaigns included absorbance of particulate matter, absorbance of dissolved organic matter, snow and sea-ice transmittance, snow/ice hyperspectral and hyperangular hemispherical-directional-reflectance (Goyens et al., 2018) and surface spectral albedo (Table 2). Downwelling spectral irradiance above the surface ($1° \times 1°$ spatial resolution, daily temporal resolution, interpolated hourly) was also computed based on the radiative transfer model SBDART (Ricchiazzi et al., 1998) as described in Laliberté et al.
(2016) and Randelhoff et al. (2019).

### 4.3   Nutrients

Nitrate, nitrite, phosphate and silicate concentrations were measured from water filtered through 0.7 µm Whatman GF/F filters and through 0.2 µm cellulose acetate membranes. Filtrates were collected into sterile 20 mL polyethylene vials, poisoned with 100 µL of mercuric chloride (60 mg L$^{-1}$) and subsequently stored in the dark prior to anal-
160 ysis. Nutrient concentrations were determined using an automated colorimetric procedure described in Aminot and Kérouel (2007). Figure 8 shows an overview of the dynamics of nitrate which is often the limiting nutrient for phytoplankton growth in the ocean (Tremblay and Gagnon, 2009). It can be seen that the depletion of the nitrates started approximately mid-June for both years, coinciding with the initiation of the phytoplankton bloom. However, the depletion was observed deeper in the water column in 2016 compared to 2015 due to stronger currents and
165 a longer sampling period in 2016 (Oziel et al., 2019). Other nutrients such as dissolved organic and inorganic carbon (DOC/DIC), particulate organic and inorganic carbon (POC/PIC), total organic carbon (TOC), phosphate (PO4), orthosilicic acid (Si(OH)$_4$), and ammonium (NH$_4$), were also measured during both campaigns (Table 2). Detailed information about analytical procedures can be found in the LEFE-CYBER online repository. A comprehensive discussion about nutrient dynamics during the Green Edge missions can be found in Grondin et al. (2019).

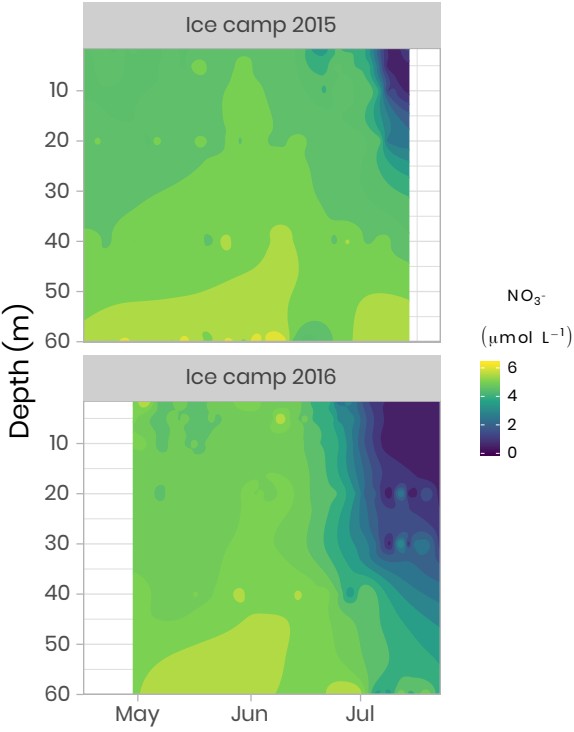

**Figure 8.** Temporal evolution of the nitrates in the first 60 m of the water column for both ice camp missions.

## 4.4 Bacteria and Phytoplankton

### 4.4.1 Flow cytometry

The abundances of pico-phytoplankton, nano-phytoplankton and bacteria were measured by flow cytometry. Samples (1.5 mL) were preserved with a mix of glutaraldehyde and Pluronic (Marie et al., 2014) and frozen at -80°C. Samples were analyzed on a FACS Canto flow cytometer (Becton Dickinson) in the laboratory at the Station Biologique de Roscoff. The abundance (cells mL$^{-1}$) of phytoplankton populations was determined on unstained samples and cells were discriminated by their red chlorophyll autofluorescence. Bacterial abundance was determined based on the fluorescence of SYBR Green-stained DNA (Marie et al., 1997). In both 2015 and 2016, bacteria concentrations were initially low, of the order of 100 000 cells mL$^{-1}$, and quite uniform throughout the water column. During the bloom, bacterial abundance increased continuously, reaching values of one million cells mL$^{-1}$ (Fig. 9). Simultaneously, the distribution of highest abundance became stratified with a higher concentration found near the surface in early July before it moved down to the subsurface (between 10 and 20 m) later in July (Fig. 9). In 2015, the sampling period did not extend long enough to capture the full progression of bacterial community development.





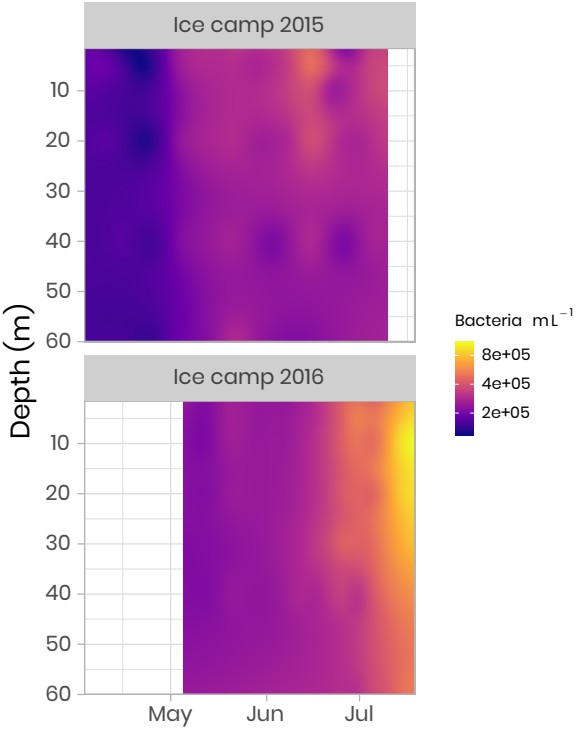

**Figure 9.** Concentration of bacteria in the water column at the ice camp in 2015 and 2016.

## 4.5 Phytoplankton

### 4.5.1 Chlorophyll a

185 Chl a and accessory pigments concentrations were determined by high-performance liquid chromatography (HPLC) following Ras2008. Concentrations were measured using volumes between 0.1 and 1 L of melted ice and volumes between 1 and 2.5 L of seawater. Water was filtered onto Whatman GF/F 25 mm filters and stored at -80°C until analysis. Filters were extracted in 100% methanol, disrupted by sonication and clarified by filtration. Pigments were analyzed using an Agilent Technologies 1200 Series system with a narrow reversed-phase C8 Zorbax Eclipse XDB

column (150 × 3 mm, 3.5 μm particle size) which was maintained at 60°C. Figure 10 shows the temporal evolution of surface integrated chl a in the bottom 10 cm of the ice cover and the water column for both years. At the beginning of the sampling periods in 2015 and 2016, total chl a concentrations in the bottom of the ice and the water column were of approximately the same magnitude ($\approx$5 mg m$^{-2}$). Later in the season, when the snowpack and the ice sheet started to melt (between June and July), and at the onset of the PSB, chl a in the water column increased rapidly to

reach concentrations of 145 mg m$^{-2}$ in 2015 and 113 mg m$^{-2}$ in 2016. At the same time, or slightly before, chl a in the ice bottom started to decrease rapidly to concentrations varying between 0.1 and 0.3 mg m$^{-2}$.

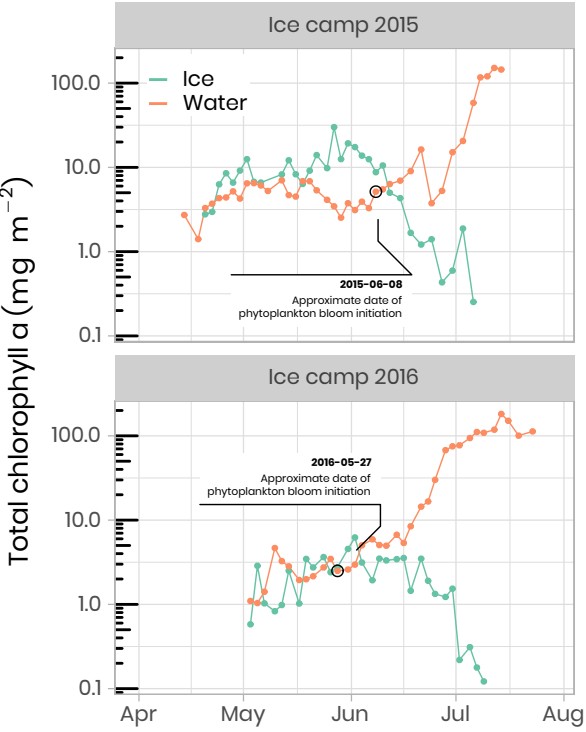

**Figure 10.** Temporal evolution of chlorophyll a in ice and water (depth-integrated) for both ice camp missions. Note that the water chlorophyll a have been integrated over the first 100 m of the water column whereas the ice chlorophyll a was measured on the bottom 0-10 cm of the ice cores. The details of the calculations to determine the approximate dates of phytoplankton bloom initiation can be found in Oziel et al. (2019).

Primary production during the phytoplankton bloom was incompletely sampled in 2015, while in 2016 it was monitored from the onset under melting sea ice in May to its termination in July (Fig. 11). During the ice-covered period in 2015, primary production, as well as nitrate assimilation (rNO$_3$), occurred at very low but detectable rates reaching 8 and 0.4 mmol m$^{-2}$ d$^{-1}$, respectively. Phytoplankton production rates were higher in the ice than in the water column, representing approximately 80% and 40% for primary production and rNO$_3$, respectively. Estimated assimilated concentrations of total carbon and nitrate within the ice cover were 30-96 and 1.4–4.6 mmol m-2 during this period. The break-up of the sea ice cover was characterized by a rapid increase in primary production and rNO$_3$. During this period of high light transmission through the melting ice cover (day 169 to 190), concentrations of assimilated total carbon and rNO$_3$ reached 60 and 8 mmol m-2, respectively, leading to a complete nitrate depletion. The quantities of total carbon and nitrate assimilated during the 2016 PSB in the water column were 562 and 97 mmol m$^{-2}$, respectively.

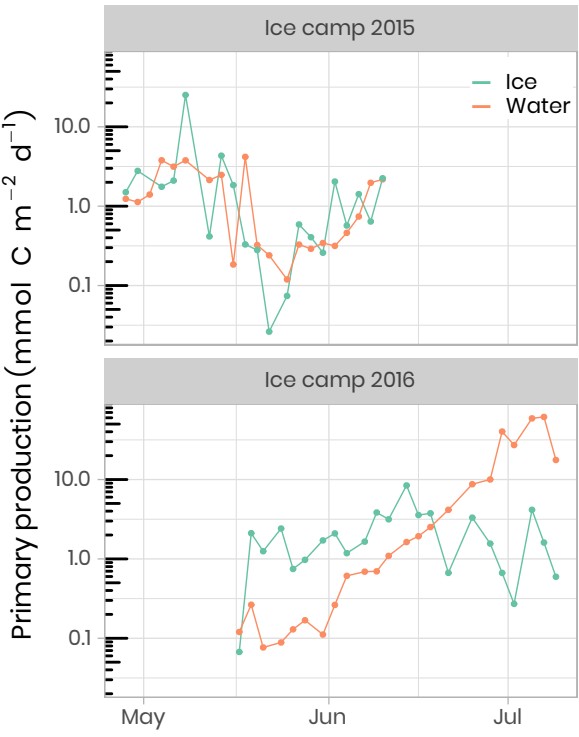

**Figure 11.** Temporal evolution of primary production a in ice and water (depth-integrated) for both ice camp missions.

### 4.5.2   Phytoplankton taxonomy

The phytoplankton community species composition was determined using an Imaging FlowCytobot (IFCB, Woods
Hole Oceanographic Institute, Sosik and Olson (2007), Olson and Sosik (2007)). The size range targeted was be-
tween 1 and 150 μm, while the image resolution of approximately 3.4 pixels μm⁻¹ limited the identification of cell
< 10 μm to broad functional groups. A 150 μm Nitex mesh was used to avoid clogging of the fluidics system by
large particles, although this might have induced a bias in the results by preventing large cells to be sampled.
For each melted ice and seawater sample, 5 mL were analyzed and Milli-Q water was run between samples with
high biomass in order to prevent contamination between samples. Image acquisition was triggered by chl a in vivo
fluorescence, with excitation and emission wavelengths of 635 and 680 nm, respectively. Grayscale images were
processed to extract regions of interest (ROIs) and their associated features (e.g.: geometry, shape, symmetry, tex-
ture, etc.), using a custom made MATLAB (2013b) code (Sosik and Olson (2007), Olson and Sosik (2007); processing
codes are available at https://github.com/hsosik/ifcb-analysis). A total of 231 features (see the full list and descrip-
tion at https://github.com/hsosik/ifcb-analysis/wiki/feature-file-documentation) were derived on the resulting ROIs
and were used for automatic classification using random forest algorithms with the EcoTaxa application (Picheral
et al., 2017). A learning set was manually prepared for each year, with ca. 20 000 images annotated and used for





automatic prediction. Each automatically annotated image was further validated by visual examination and corrected when necessary. The final 2015 and 2016 datasets consist of 124 247 and 57 397 annotated images and
their associated features in 39 and 35 taxonomic categories, respectively (Fig. 12). As it was impossible to count the number of cells in each image, we assumed one cell per image. To account for potential underestimations of cell abundance when colonies or chains were imaged, the biovolume of each living protist on images was computed during image processing according to Moberg and Sosik (2012). Using carbon to volume ratios from Menden-Deuer and Lessard (2000), biovolume was converted into carbon estimates, as described in Laney and Sosik (2014). De-
tailed information about sea ice algae and phytoplankton community composition can be found in Grondin et al. (2019).



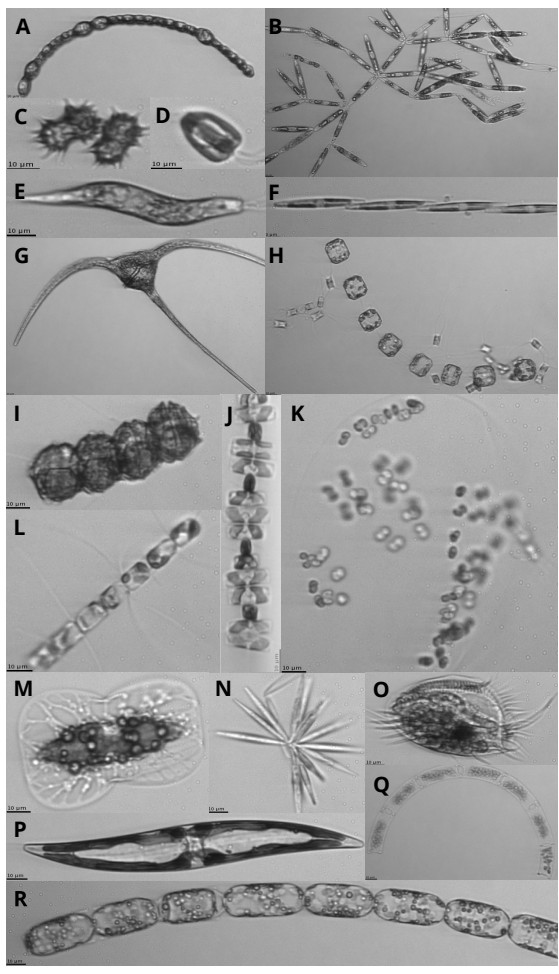

**Figure 12.** Images of protists sampled with the IFCB. Scale bar on images is 10 µm. Note that images are not to scale. (**A**) *Anabaena* sp. (**B**) *Nitzschia frigida* (**C**) *Polarella glacialis* (**D**) Flagellate (**E**) Euglena (**F**) *Pseudo-nitzschia* sp. (**G**) *Ceratium* sp. (**H**) *Thalassiosira nordenskioeldii* with *Attheya septentrionalis* (**I**) *Peridiniella catenata* (**J**) *Navicula pelagica* (**K**) *Phaeocystis* sp. colony (**L**) *Chaetoceros* sp. (**M**) *Entomoneis* sp. (**N**) *Synedropsis hyperborea* (**O**) Ciliate (**P**) Pennate diatom (**Q**) *Eucampia* sp. (**R**) *Melosira* sp.

### 4.5.3 Physiology of the phytoplankton community

The photosynthetic potential of microalgae was assessed by measuring $Fv/Fm$, namely the maximum photochemical efficiency of Photosystem II (PSII), via dynamic chl a fluorescence:

$$\frac{Fv}{Fm} = \frac{(Fm - F0)}{Fm} \tag{1}$$

where $Fm$ and $F0$ are the maximum and minimum PSII chl a fluorescence yields, respectively. Chl a fluorescence was recorded with a Water-PAM fluorometer (Walz, Germany) on melted sea-ice (last centimeter of the cores) and



water samples collected at different depths (i.e. 1.5 m, 10 m, 40 m, 60 m). Measurements were performed after storing samples in 50 mL dark Falcon tubes (Corning Life Sciences, USA) on ice for at least 1 h. For further technical

details, see Galindo et al. (2017). $Fv/Fm$ is often used as an index for evaluating the physiological condition of microalgal communities. For algae that are growing optimally, the $Fv/Fm$ ratio ranges between 0.50 and 0.75 in the absence of cyanobacteria. Below 0.50, algal growth is considered to be limited by nutrient availability and/or light stress (Suggett et al., 2010). Figure 13 shows the temporal evolution of $Fv/Fm$ for ice algae and phytoplankton for the ice camp in 2016. At the beginning of the sampling period, all samples showed $Fv/Fm$ above 0.55. While

in ice $Fv/Fm$ ranged between 0.60 and 0.75 until the beginning of June, it decreased to ca. 0.20-0.35 in water. This decrease of $Fv/Fm$ (Fig. 13A) is coincident with a sharp increase in PAR under the ice sheet (Fig. 5), which may have induced light stress in phytoplankton and ice algae communities. After approximately 1 month, phytoplankton became acclimated to this new light environment and $Fv/Fm$ increased back to 0.60-0.75 by the beginning of June. From that time on (corresponding to higher irradiance transmittance through ice, see Fig. 5), $Fv/Fm$ in ice

decreased dramatically to an approximate value of 0.20 while $Fv/Fm$ in the water column generally remained between 0.60 and 0.75 for depths between 10 and 60 m (note however the large decrease at 40 m on June 13). In contrast, $Fv/Fm$ at 1.5 m was lower and noisier with values varying between 0.45 and 0.60.





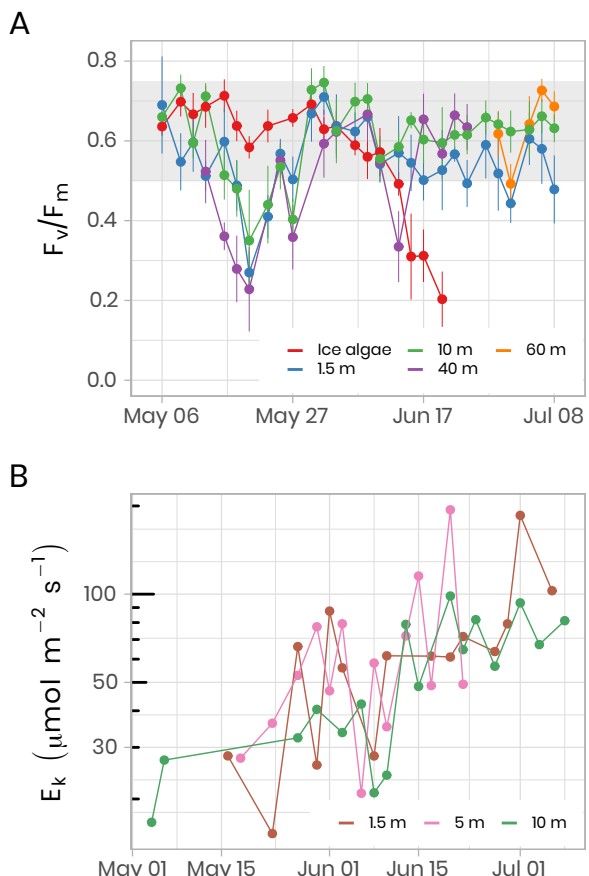

**Figure 13. (A)** Temporal evolution of $F_v/F_m$ for ice (last cm) and water underneath the ice (depths 1.5 m, 10 m, 40 m) samples for the ice camp 2016 between May 6[th] and July 8[th]. $F_v/F_m$ monitoring on ice samples stopped on June 20th because the chl a fluorescence signal was not reliable anymore. $F_v/F_m$ monitoring on 40 m and 60 m depth samples was limited between May 13th and June 24th and between June 29th and July 08th, respectively. The gray shaded area represents the range at which the algae are optimally growing. **(B)** The light saturation parameter, $E_k$, an index of photoadaptation of the phytoplankton community measured at 1.5 m, 5 m and 10 m depth. Note de log scale on the $y$ axis.

In addition to the photosynthetic potential of microalgae, photosynthetic parameters were measured from seawater incubated at different irradiance levels in the presence of [14]C labelled sodium bicarbonate. The light saturation parameter, $E_k$, is an indication of the physiological state of the phytoplankton community. Figure 13B shows the increase of $E_k$ as the phytoplankton community grows between May and July of 2016 at 1.5 m, 5 m and 10 m depth. Between 1.5 m and 10 m depth, $E_k$ varied between 15 and 194 µmol m[-2] s[-1] (61 ± 37 µmol m[-2] s[-1], $n$ = 69) which fall in range within values reported in other marine studies conducted at high-latitudes (Bouman et al., 2018; Massicotte et al., 2019b). The observed increase in $E_k$ over the growing season suggests that the phytoplankton community became more photo-adapted to increasing available irradiance (Fig. 5).



### 4.6 Zooplankton

Zooplankton was collected from a ring net deployed under the ice at the ice camp between April 22 and June 10 in 2015 and between May 16 and July 18 in 2016. This sampler, composed of a 1 m diameter circular frame mounted with a 4 m long 200 μm mesh size conical plankton net was lowered cod-end first to avoid filtration during the descent, using an electric winch. An additional 50 μm net with an aperture of 10 cm in diameter was attached to the side of the metal ring to sample eggs and small zooplankton larvae while the main net collected the mesozooplankton fraction. This sampling device was hauled vertically from a depth of 100 m (2015 and 2016) or 350 m (only in 2016), 10 m above the seafloor to the surface at a speed of about 30m min-1. The filtered volume was estimated by a KC Denmark flowmeter placed in the mouth of the 200 μm mesh net. Samples were preserved in 10% buffered formalin seawater solution for further taxonomic analyses. Classification and count of the 200 μm mesh net samples from both campaigns were performed using the zooscan by the PIQv team at l'Observatoire Océanographique de Villefranchesur-Mer, France, following their protocol. Figure 14 shows the time series of the abundance of copepods (the dominant group of zooplankton in the Arctic) for the first 100 m and 350 m of the water column in 2016.

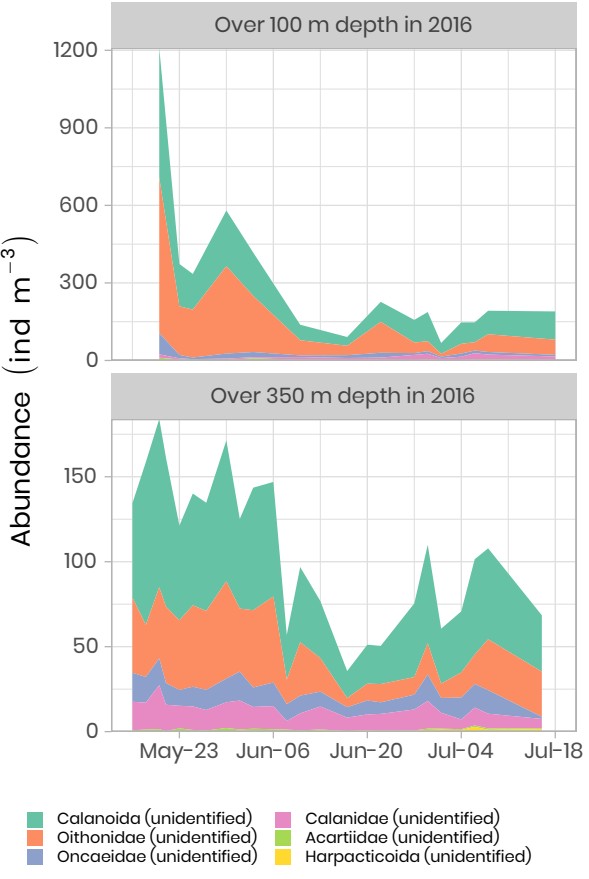

**Figure 14.** Time series of the abundance of the copepods (ind m$^{-3}$) measured over the first 100 m and 350 m of the water column in 2016 using the zooscan. For visualization, only the six most abundant groups are presented in decreasing order of importance. Note the different $y$ axes in both panels.

Highest copepod abundance was observed in late May and early June in both the top 100 m and over 350 m hauling depths. At the beginning of the sampling period, abundance was approximately 10 times higher in the first 100 m of the water column than over 350 m, suggesting that copepods were agglomerating near the surface to exploit the ice algae production before the start of phytoplankton production. Abundance started to decrease during the first week of June. The family of Oithonidae and the order of Calanoida were the two most abundant groups

over the 2 sampling depths. Oithonidae was more abundant over the top 100m layer as this group is probably mainly composed of small epipelagic *Oithona similis* one of the most numerous copepods in the Arctic. Calanoida, the most common copepod order, which includes the families Calanidae (including species such as *Calanus spp.*) and Acartiidae, was the dominant group over the 350m depth haul.





### 4.7 Other data

An exhaustive list of all measured variables is presented in Table 2 along with contact information of principal investigators associated with each measured parameter.

### 5 Recommendations and lessons learned

As with any Arctic surveys, a large number of measurements were acquired during the Green Edge project. Although initial recommendations on good practices about collection, processing and storage of collected data were commu-
290 nicated to all scientists, extensive efforts, such as data standardization, had to be performed to assemble the data. It is important for reducing possible errors, that a uniformized data management plan should be prepared and distributed prior to each mission. Furthermore, dedicated data management specialists should be involved from the beginning of the project to ensure the data are adequately collected, tidied, stored, backed up and archived.

### 6 Conclusions

The comprehensive data set assembled during both Green Edge ice-camp campaigns allowed us to study the fundamental physical, chemical and biological processes controlling the Arctic PSB. In this paper, only a handful of variables have been presented. The reader can find the complete list of measured variables in Table 2, all of which are also fully available in the data repository. Furthermore, a collection of scientific research papers is currently being submitted to a special issue of the Elementa journal entitled *Green Edge -The phytoplankton spring bloom in*
*the Arctic Ocean: past, present and future response to climate variations, and impact on carbon fluxes and the marine food web*. The uniqueness and comprehensiveness of this data set offer more opportunities to reuse it for other applications.

### 7 Code and data availability

The raw data provided by all the researchers, as well as metadata, are available on the LEFE-CYBER repository
(http://www.obs-vlfr.fr/proof/php/GREENEDGE/greenedge.php). The data presented in this paper and in Table 2 are hosted at SEANOE (SEA scieNtific Open data Edition) under the CC-BY license (https://www.seanoe.org/data/00487/59892/, Massicotte et al. (2019a)). Detailed metadata are associated with each file including the principal investigator's contact information. For specific questions, please contact the principal investigator associated with the data (see Table 2).

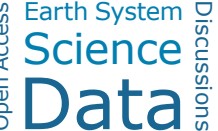

# Tables

Table 1: Descriptions of the minimal variables included in each data set (i.e. in each CSV file).

| Variable | Description |
|---|---|
| date | Sampling date (UTC) |
| latitude | Latitude of the sampling location (degree decimals). |
| longitude | Longitude of the sampling location (degree decimals). |
| sample_type | Origin of the water ("water", "ice", "meltpond"). |
| sample_source | Source of the water ("niskin", "underice" "0-1 cm", "0-3 cm", "3-10 cm", "rosette" ). |
| depth_m | Depth at which measurement was made. |
| snow_thickness | Qualitative value describing the snow cover under which measurement was made ("thin_snow", "thick_snow"). |
| mission | Mission identifier ("ice_camp_2015", "ice_camp_2016") |
| pi | Name(s) of the principal investigator(s) responsible of the measured variable. |



Table 2: Parameters measured during the Green Edge ice camp surveys. Parameters are ordered by alphabetical order and sampling year.

| Year | Parameter | Sampling method | Principal investigators | Processed |
|------|-----------|-----------------|-------------------------|-----------|
| 2015 | Absorption coefficient | In-water profiler | Becu G. / Babin M. | Available |
| 2015 | Absorption (particulate) | Camp ice sample | Ehn J. / Cox C. | Available |
| 2015 | Absorption (particulate) | Camp water sample | Ehn J. / Cox C. | Available |
| 2015 | Absorption (particulate) | Camp ice sample | Matsuoka A. / Bricaud A. / Ferland J. | Available |
| 2015 | Absorption (particulate) | Camp water sample | Matsuoka A. / Bricaud A. / Ferland J. | Available |
| 2015 | ADCP (Mooring) | Mooring | Marec C. | Available |
| 2015 | Aerosol optical depth | Surface mode | Belanger S. / Goyens C. / Leymarie E. | Available |
| 2015 | Aerosol relative humidity | Surface mode | Belanger S. / Goyens C. / Leymarie E. | Available |
| 2015 | Air Relative Humidity | Meteorological Tower | Massé G. | Available |
| 2015 | Air Temperature | Meteorological Tower | Massé G. | Available |
| 2015 | Alkalinity total (TA) | Camp water sample | Else B. / Whitehead J. | Available |
| 2015 | Ammonium ($NH_4^+$) | Camp water sample | Raimbault P. | Data not available yet |
| 2015 | Ammonium ($NH_4^+$, assimilation) | Camp water sample | Raimbault P. | Available |
| 2015 | Ammonium ($NH_4^+$, regeneration) | Camp water sample | Raimbault P. | Available |
| 2015 | Angstrom coefficient | Surface mode | Belanger S. / Goyens C. / Leymarie E. | Available |
| 2015 | Attenuation coefficient | In-water profiler | Becu G. / Babin M. | Available |
| 2015 | Backscattering coefficient | In-water profiler | Becu G. / Babin M. | Available |
| 2015 | Bacterial sequencing | Air filtration | Amiraux R. | Available |
| 2015 | Bacterial sequencing | Camp water sample | Amiraux R. | Available |
| 2015 | Bacterial sequencing | Ice core | Amiraux R. | Available |
| 2015 | Bacterial sequencing | Sediment trap | Amiraux R. | Available |
| 2015 | Brine salinity and volume | Sea ice core | Galindo V./ Rysgaard S. | Available |
| 2015 | Chlorophyll a | In-water profiler | Becu G. / Babin M. | Available |
| 2015 | Chlorophyll a | Sediment Trap | Fortier L. / Lalande C. | Available |
| 2015 | Chlorophyll a and Phaeopigments (concentration) | Camp water sample | Babin M. / Ferland J. | Available |
| 2015 | Chlorophyll a and phaeopigments (concentration) | Camp water sample | Raimbault P. | Data not available yet |
| 2015 | Chromophoric dissolved organic matter absorption | In-water profiler | Becu G. / Babin M. | Available |
| 2015 | Chromophoric dissolved organic matter absorption | Camp water sample | Matsuoka A. / Ferland J. / Babin M. | Available |
| 2015 | Conductivity, temperature, and depth (CTD) | In-water profiler | Becu G. / Babin M. | Available |
| 2015 | Conductivity, temperature, and depth (CTD) | In-water profiler | Guillot P. / Babin M. / Marec C. | Available |
| 2015 | Cryptophytes (abundance) | Camp water sample | Vaulot D. / Marie D. | Available |

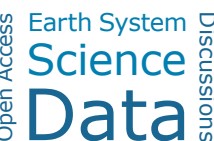
Table 2: Parameters measured during the Green Edge ice camp surveys. Parameters are ordered by alphabetical order and sampling year. *(continued)*

| Year | Parameter | Sampling method | Principal investigators | Processed |
|------|-----------|-----------------|-------------------------|-----------|
| 2015 | Diffuse attenuation coefficient (Kd) | Profile mode | Becu G. / Babin M. | Available |
| 2015 | Dimethyl sulfide (DMS) | Camp water sample | Levasseur M. | Available |
| 2015 | Dimethyl sulfide (DMS) | Melt pond water sample | Levasseur M. | Available |
| 2015 | Dimethyl sulfide (DMS) | Sea ice core | Levasseur M. | Available |
| 2015 | Dimethylsulfoniopropionate (DMSP) | Camp water sample | Levasseur M. | Available |
| 2015 | Dimethylsulfoniopropionate (DMSP) | Melt pond water sample | Levasseur M. | Available |
| 2015 | Dimethylsulfoniopropionate (DMSP) | Sea ice core | Levasseur M. | Available |
| 2015 | Dissolved inorganic Carbon (DIC) | Camp water sample | Else B. / Whitehead J. | Available |
| 2015 | Dissolved organic matter (sugars) | Rosette | Sempéré R. / Panagiotopoulos C. | Available |
| 2015 | Dissolved organic nitrogen (release) | Camp water sample | Raimbault P. | Available |
| 2015 | Downwelling irradiance | Surface mode | Belanger S. / Goyens C. / Leymarie E. | Available |
| 2015 | Downwelling Irradiance above the surface ($E_d(0^+)$) | Surface mode | Babin M. / Galí M. | Available |
| 2015 | Downwelling Irradiance above the surface ($E_d(0^+)$) | Profile mode | Becu G. / Babin M. | Available |
| 2015 | Downwelling Irradiance ($E_d(z)$) | Profile mode | Becu G. / Babin M. | Available |
| 2015 | $E_d(0^+)$ spectra from SBDART radiative transfer simulations | Surface mode | Babin M. / Galí M. | Available |
| 2015 | Faecal pellets flux | Sediment Trap | Fortier L. / Lalande C. | Available |
| 2015 | Fluorescence Variable (phytoplankton) | Camp water sample | Galindo V. / Rysgaard S. | Data not available yet |
| 2015 | Fluorescence Variable (phytoplankton) | Sediment Trap | Galindo V. / Rysgaard S. | Available |
| 2015 | Fluorescence Variable (phytoplankton) | Surface mode | Galindo V. / Rysgaard S. | Data not available yet |
| 2015 | Hemispherical directional reflectance distribution function | Surface mode | Belanger S. / Goyens C. / Leymarie E. | Available |
| 2015 | Hemispherical Directional Reflectance Factor | Surface mode | Belanger S. / Goyens C. / Leymarie E. | Available |
| 2015 | Heterotrophic bacteria (abundance) | Camp water sample | Vaulot D. / Marie D. | Available |
| 2015 | Heterotrophic nanoflagellates | Camp water sample | Joux F. | Available |
| 2015 | Ice and snow temperature | Meteorological Tower | Massé G. | Available |
| 2015 | Ice thickness | Camp ice sample | Galindo V. / Rysgaard S. | Available |
| 2015 | Irradiance (downwelling, upwelling) | Surface-, Under-water profile-mode | Matthes L. / Ehn J. / Lambert-Girard S./ Mundy C.J. | Available |
| 2015 | Isoprenoid lipids | Camp water sample | Massé G. / Guilmette C. | Data not available yet |
| 2015 | Isoprenoid lipids | Sea ice core | Massé G. / Guilmette C. | Data not available yet |
| 2015 | Net radiation | Surface mode | Else B. | Available |
| 2015 | Nitrate ($NO_3^-$) | Camp water sample | Raimbault P. | Available |
| 2015 | Nitrate ($NO_3^-$) | Sea ice core | Raimbault P. | Available |
| 2015 | Nitrate ($NO_3^-$, assimilation) | Camp water sample | Raimbault P. | Available |

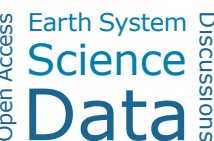

Table 2: Parameters measured during the Green Edge ice camp surveys. Parameters are ordered by alphabetical order and sampling year. *(continued)*

| Year | Parameter | Sampling method | Principal investigators | Processed |
|------|-----------|-----------------|-------------------------|-----------|
| 2015 | Nitrification | Camp water sample | Raimbault P. | Available |
| 2015 | Nitrite ($NO_2^-$) | Camp water sample | Raimbault P. | Available |
| 2015 | Nitrite ($NO_2^-$) | Sea ice core | Raimbault P. | Available |
| 2015 | PAR from SBDART radiative transfer simulations | Surface mode | Babin M. / Galí M. | Available |
| 2015 | Particle Size Distribution | In-water profiler | Becu G. / Babin M. | Available |
| 2015 | Particles size | Underwater Vision Profiler (UVP) | Marec C. / Picheral M. | Available |
| 2015 | Particulate Carbon (PC) | Camp water sample | Babin M. / Ferland J. | Available |
| 2015 | Particulate mass | Sediment Trap | Fortier L. / Lalande C. | Available |
| 2015 | Particulate Nitrogen (PN) | Camp water sample | Babin M. / Ferland J. | Available |
| 2015 | Particulate nitrogen (PN) | Sediment Trap | Fortier L. / Lalande C. | Data not available yet |
| 2015 | Particulate organic carbon (POC) | Sediment Trap | Fortier L. / Lalande C. | Available |
| 2015 | Particulate organic carbon (POC) | Camp water sample | Raimbault P. | Available |
| 2015 | Particulate organic nitrogen (PON) | Camp water sample | Raimbault P. | Available |
| 2015 | Particulate Organic Phosphorus (POP) | Camp water sample | Raimbault P. | Data not available yet |
| 2015 | PDMPO uptake | Camp water sample | Leynaert A. | Data not available yet |
| 2015 | PDMPO uptake per species | Camp water sample | Leynaert A. | Data not available yet |
| 2015 | Phosphate (($PO_4$)$^{3-}$) | Camp water sample | Raimbault P. | Available |
| 2015 | Phosphate (($PO_4$)$^{3-}$) | Sea ice core | Raimbault P. | Available |
| 2015 | Photosynthetically available radiation (PAR) | Surface mode | Babin M. / Galí M. | Available |
| 2015 | Photosynthetically available radiation (PAR) | Profile mode | Becu G. / Babin M. | Available |
| 2015 | Photosynthetic nanoeukaryotes (abundance) | Camp water sample | Vaulot D. / Marie D. | Available |
| 2015 | Photosynthetic parameters | Camp water sample | Ferland J. / Babin M. | Available |
| 2015 | Photosynthetic picoeukaryotes (abundance) | Camp water sample | Vaulot D. / Marie D. | Available |
| 2015 | Phytoplankton | Camp water sample | Ferland J. / Grondin P.L. / Babin M. / Marec C. | Available |
| 2015 | Phytoplankton (taxonomy) | Sediment Trap | Fortier L. / Lalande C. | Available |
| 2015 | Pigments | Sea ice core | Galindo V. / Rysgaard S. | Available |
| 2015 | Pigments | Camp water sample | Ras J. / Claustre H. | Available |
| 2015 | Primary production | Camp water sample | Raimbault P. | Available |
| 2015 | Rrs ($0^+$) | Profile mode | Becu G. / Babin M. | Available |
| 2015 | Salinity | Sea ice core | Galindo V. / Rysgaard S. | Available |
| 2015 | Salinity-induced bacterial biomarker | Ice core | Amiraux R./ Rontani J-F. | Available |
| 2015 | Salinity-induced bacterial biomarker | Sediment trap | Amiraux R./ Rontani J-F. | Available |

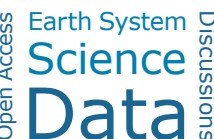

Table 2: Parameters measured during the Green Edge ice camp surveys. Parameters are ordered by alphabetical order and sampling year. *(continued)*

| Year | Parameter | Sampling method | Principal investigators | Processed |
|------|-----------|-----------------|------------------------|-----------|
| 2015 | Sea ice concentration | Surface mode | Massicotte P. | Available |
| 2015 | Silica Biogenic (BSi) | Camp water sample | Leynaert A. | Available |
| 2015 | Silica Biogenic (BSi) dissolution rate | Camp water sample | Leynaert A. | Available |
| 2015 | Silicate $Si(OH)_4$ - absorption kinetics | Camp water sample | Leynaert A. | Available |
| 2015 | Silica (uptake rate) | Camp water sample | Leynaert A. | Available |
| 2015 | $Si(OH)_4$ | Camp water sample | Leynaert A. | Available |
| 2015 | $Si(OH)_4$ | Camp water sample | Raimbault P. | Available |
| 2015 | $Si(OH)_4$ | Sea ice core | Raimbault P. | Available |
| 2015 | Snow depth | Camp snow sample | Galindo V./ Rysgaard S. | Available |
| 2015 | Snow depth | Meteorological Tower | Massé G. | Available |
| 2015 | Sugars | Sediment Trap | Sempéré R. / Panagiotopoulos C. | Available |
| 2015 | Surface Albedo | Surface mode | Verin G. | Available |
| 2015 | Suspended particulate material (SPM) | Camp water sample | Babin M. / Ferland J. | Available |
| 2015 | Swimmers | Sediment Trap | Fortier L. / Lalande C. | Available |
| 2015 | Synechococcus (abundance) | Camp water sample | Vaulot D. / Marie D. | Available |
| 2015 | Temperature | Sea ice core | Galindo V. / Rysgaard S. | Available |
| 2015 | Total organic carbon (TOC) | Rosette | Sempéré R. / Panagiotopoulos C. | Available |
| 2015 | Total organic carbon (TOC) | Camp water sample | Raimbault P. | Available |
| 2015 | Total organic nitrogen (TON) | Camp water sample | Raimbault P. | Available |
| 2015 | Total organic phosphorus (TOP) | Camp water sample | Raimbault P. | Available |
| 2015 | Transmittance through ice | Surface mode | Verin G. | Available |
| 2015 | Under-ice export fluxes of biogenic matter (fresh) | Sediment Trap | Fortier L. / Lalande C. | Available |
| 2015 | Under-ice photos and video | GoPro Hero 4 on radiometer profiler | Rehm E. | Available |
| 2015 | Upwelling Irradiance ($E_u(z)$) | Profile mode | Becu G. / Babin M. | Available |
| 2015 | Upwelling radiance ($L_u(z)$) | Surface mode | Belanger S. / Goyens C. / Leymarie E. | Available |
| 2015 | Upwelling radiance ($L_u(z)$) | Profile mode | Becu G. / Babin M. | Available |
| 2015 | Vertical profile of snow density | Surface mode | Verin G. | Available |
| 2015 | Vertical profile of Specific Surface Area | Surface mode | Verin G. | Available |
| 2015 | Virus (abundance) | Camp water sample | Joux F. | Available |
| 2015 | Wind Direction | Meteorological Tower | Massé G. | Available |
| 2015 | Wind Speed | Meteorological Tower | Massé G. | Available |





Table 2: Parameters measured during the Green Edge ice camp surveys. Parameters are ordered by alphabetical order and sampling year. *(continued)*

| Year | Parameter | Sampling method | Principal investigators | Processed |
|---|---|---|---|---|
| 2015 | Zooplancton (Abundances) | Plankton net | Fortier L. / Aubry C | Available |
| 2015 | Zooplancton (Abundances) | Plankton net (LOKI) | Fortier L. / Aubry C | Available |
| 2015 | Zooplancton (Taxonomy) | Plankton Net | Fortier L. / Aubry C | Available |
| 2015 | Zooplancton (Taxonomy) | Plankton net (LOKI) | Fortier L. / Aubry C | Available |
| 2015 | Zooplankton vertical distribution | Underwater Vision Profiler (UVP) | Marec C. / Sophie R. / Picheral M. | Available |
| 2016 | 234Th (dissolved) | Rosette | Schmidt S. | Data not available yet |
| 2016 | 234Th (particulate) | Rosette | Schmidt S. | Data not available yet |
| 2016 | Absorption coefficient | In-water IOP profiler | Becu G. / Babin M. | Available |
| 2016 | Absorption (particulate) | Camp ice sample | Matsuoka A. / Bricaud A. / Ferland J. | Available |
| 2016 | Absorption (particulate) | Camp water sample | Matsuoka A. / Bricaud A. / Ferland J. | Available |
| 2016 | ADCP (Mooring) | Mooring | Oziel L. / Houssais M.-N. / Babin M./ Lagunas J. | Available |
| 2016 | Air Relative Humidity | Meteorological Tower | Massé G. | Available |
| 2016 | Air Temperature | Meteorological Tower | Massé G. | Available |
| 2016 | Ammonium ($NH_4^+$) | Camp water sample | Raimbault P. | Data not available yet |
| 2016 | Ammonium ($NH_4^+$, assimilation) | Camp water sample | Raimbault P. | Available |
| 2016 | Ammonium ($NH_4^+$, regeneration) | Camp water sample | Raimbault P. | Available |
| 2016 | Attenuation coefficient | In-water IOP profiler | Becu G. / Babin M. | Available |
| 2016 | Backscattering coefficient | In-water IOP profiler | Becu G. / Babin M. | Available |
| 2016 | Bacterial cultures | Camp water sample | Joux F. | Available |
| 2016 | Bacterial cultures | Sea ice core | Joux F. | Available |
| 2016 | Bacterial production | Sea ice core | Joux F. / Galindo V. | Available |
| 2016 | Bacterial production | Camp water sample | Joux F. / Galindo V. | Available |
| 2016 | Brine salinity and volume | Sea ice core | Galindo V./ Rysgaard S. | Available |
| 2016 | Chlorophyll a | In-water IOP profiler | Becu G. / Babin M. | Available |
| 2016 | Chlorophyll a | Sediment Trap | Fortier L. / Lalande C. | Available |
| 2016 | Chlorophyll a and phaeopigments (concentration) | Camp water sample | Babin M. / Ferland J. | Available |
| 2016 | Chromophoric dissolved organic matter absorption | In-water IOP profiler | Becu G. / Babin M. | Available |
| 2016 | Chromophoric dissolved organic matter absorption | Camp water sample | Matsuoka A. / Ferland J. / Babin M. | Available |
| 2016 | Chromophoric dissolved organic matter fluorescence | Camp water sample | Matsuoka A. / Ferland J. | Available |
| 2016 | Conductivity, temperature, and depth (CTD) | In-water IOP profiler | Becu G. / Babin M. | Available |
| 2016 | Conductivity, temperature, and depth (CTD) | In-water profiler | Guillot P. / Lagunas J. | Available |
| 2016 | Cryptophytes (abundance) | Camp water sample | Vaulot D. | Available |





Table 2: Parameters measured during the Green Edge ice camp surveys. Parameters are ordered by alphabetical order and sampling year. *(continued)*

| Year | Parameter | Sampling method | Principal investigators | Processed |
|------|-----------|-----------------|-------------------------|-----------|
| 2016 | Cultures of sorted populations | Camp water sample | Vaulot D. | Available |
| 2016 | Diatoms (bacilliarophyta) abundance | Camp water sample | Leblanc K. / Queguiner B. / Lafond.A | Available |
| 2016 | Diatoms (bacilliarophyta) taxonomy | Camp water sample | Leblanc K. / Queguiner B. / Lafond.A | Available |
| 2016 | Diffuse attenuation coefficient (Kd) | Optical radiometers profiling system | Becu G. / Babin M. | Available |
| 2016 | Dissolved organic carbon (HTCO) | Rosette | Matsuoka A. / Benner R. / Ferland J. | Available |
| 2016 | Dissolved organic matter (Amino acids) | Rosette | Matsuoka A. / Benner R. / Ferland J. | Available |
| 2016 | Dissolved organic matter (sugars) | Rosette | Panagiotopoulos C./ R Sempéré | Available |
| 2016 | Dissolved organic nitrogen (release) | Camp water sample | Raimbault P. | Available |
| 2016 | Downwelling irradiance | Surface mode | Belanger S. / Goyens C. / Lambert Girard S. | Available |
| 2016 | Downwelling irradiance | Surface mode | Lambert-Girard S. / Leymarie E. | Available |
| 2016 | Downwelling Irradiance above the surface ($E_d(0^+)$) | Surface mode | Babin M. / Galí M. | Available |
| 2016 | Downwelling Irradiance above the surface ($E_d(0^+)$) | Optical radiometers profiling system | Becu G. / Babin M. | Available |
| 2016 | Downwelling Irradiance ($E_d(z)$) | Optical radiometers profiling system | Becu G. / Babin M. | Available |
| 2016 | $E_d(0^+)$ spectra from SBDART radiative transfer simulations | Surface mode | Babin M. / Galí M. | Available |
| 2016 | Faecal pellets flux | Sediment Trap | Fortier L. / Lalande C. | Available |
| 2016 | Hemispherical directional reflectance distribution function | Surface mode | Belanger S. / Goyens C. / Lambert-Girard S. | Data not available yet |
| 2016 | Heterotrophic bacteria (abundance) | Camp water sample | Vaulot D. | Available |
| 2016 | Hydro SCAMP (temperature, salinity, chlorophyll, turbidity, etc.) | In-water profiler | Vladoiu A. / Dumont D. / Sévigny C. | Available |
| 2016 | Ice and snow temperature | Meteorological Tower | Massé G. | Data not available yet |
| 2016 | Ice thickness | Camp ice sample | Galindo V./ Rysgaard S. | Available |
| 2016 | Irradiance (downwelling) | Surface-, Ice Bottom-mode | Matthes L. / Ehn J. / Lambert-Girard S./ Mundy C.J. | Available |
| 2016 | Irradiance (downwelling) | Under-ice irradiance transects, ROV | Matthes L. /Lambert-Girard S./ Ehn J./Mundy C.J | Available |
| 2016 | Irradiance (downwelling, upwelling) | Surface-, Under-water profile-mode | Matthes L. / Ehn J. / Lambert-Girard S./ Mundy C.J. | Available |
| 2016 | Isoprenoid lipids | Camp water sample | Massé G. / Guilmette C. | Data not available yet |
| 2016 | Isoprenoid lipids | Sea ice core | Massé G. / Guilmette C. | Data not available yet |
| 2016 | Lipid biomarkers | Collected organisms | Dufour F. / Massé G. / Ayotte P. / Lemire M. | Data not available yet |
| 2016 | Lipid tracers of bacteria stress | Camp water sample | Rontani J.-F. / Amiraux R. | Data not available yet |
| 2016 | Lipid tracers of bacteria stress | Sea ice core | Rontani J.-F. / Amiraux R. | Data not available yet |
| 2016 | Lipid tracers of bacteria stress | Sediment Trap | Rontani J.-F. / Amiraux R. | Data not available yet |
| 2016 | Nitrate ($NO_3^-$) | Camp water sample | Raimbault P. | Available |
| 2016 | Nitrate ($NO_3^-$) | Sea ice core | Raimbault P. | Available |
| 2016 | Nitrate ($NO_3^-$, assimilation) | Camp water sample | Raimbault P. | Available |



Table 2: Parameters measured during the Green Edge ice camp surveys. Parameters are ordered by alphabetical order and sampling year. *(continued)*

| Year | Parameter | Sampling method | Principal investigators | Processed |
|------|-----------|-----------------|-------------------------|-----------|
| 2016 | Nitrification | Camp water sample | Raimbault P. | Available |
| 2016 | Nitrite ($NO_2^-$) | Camp water sample | Raimbault P. | Available |
| 2016 | Nitrite ($NO_2^-$) | Sea ice core | Raimbault P. | Available |
| 2016 | Nutrients bioassay | Experiment | Delaforge A./ Mundy CJ | Data not available yet |
| 2016 | Nutrients bioassay | Experiment | Galindo V./ Rysgaard S. | Data not available yet |
| 2016 | PAR from SBDART radiative transfer simulations | Surface mode | Babin M. / Galí M. | Available |
| 2016 | Particle Size Distribution | In-water IOP profiler | Becu G. / Babin M. | Data not available yet |
| 2016 | Particle Size Distribution | In-water profiler | L. Stemmann / Lagunas J. | Data not available yet |
| 2016 | Particles size | Underwater Vision Profiler (UVP) | Lagunas J. / Picheral M. | Available |
| 2016 | Particulate Carbon (PC) | Camp water sample | Babin M. / Ferland J. | Available |
| 2016 | Particulate mass | Sediment Trap | Fortier L. / Lalande C. | Available |
| 2016 | Particulate Nitrogen (PN) | Camp water sample | Babin M. / Ferland J. | Available |
| 2016 | Particulate nitrogen (PN) | Sediment Trap | Fortier L. / Lalande C. | Available |
| 2016 | Particulate organic carbon (POC) | Sediment Trap | Fortier L. / Lalande C. | Available |
| 2016 | Particulate organic carbon (POC) | Camp water sample | Raimbault P. | Available |
| 2016 | Particulate organic nitrogen (PON) | Camp water sample | Raimbault P. | Available |
| 2016 | Particulate Organic Phosphorus (POP) | Camp water sample | Raimbault P. | Data not available yet |
| 2016 | PDMPO uptake | Camp water sample | Leblanc K. / Queguiner B. | Data not available yet |
| 2016 | PDMPO uptake per species | Camp water sample | Leblanc K. / Queguiner B. | Data not available yet |
| 2016 | Phosphate (($PO_4$)$^{3-}$) | Camp water sample | Raimbault P. | Available |
| 2016 | Phosphate (($PO_4$)$^{3-}$) | Sea ice core | Raimbault P. | Available |
| 2016 | Photosynthetically available radiation (PAR) | Surface mode | Babin M. / Galí M. | Available |
| 2016 | Photosynthetically available radiation (PAR) | Optical radiometers profiling system | Becu G. / Babin M. | Available |
| 2016 | Photosynthetic eukaryotes (morphology) | Camp water sample | Vaulot D. | Available |
| 2016 | Photosynthetic nanoeukaryotes (abundance) | Camp water sample | Vaulot D. | Available |
| 2016 | Photosynthetic parameters | Camp water sample | Ferland J. / Babin M. | Available |
| 2016 | Photosynthetic parameters (variable fluorescence) | Camp water sample | Lavaud J. / Galindo V. / Rysgaard S. | Data not available yet |
| 2016 | Photosynthetic parameters (variable fluorescence) | Sediment Trap | Lavaud J. / Galindo V. / Rysgaard S. | Data not available yet |
| 2016 | Photosynthetic parameters (variable fluorescence) | Sea ice core | Lavaud J. / Galindo V. / Rysgaard S. | Data not available yet |
| 2016 | Photosynthetic picoeukaryotes (abundance) | Camp water sample | Vaulot D. | Available |
| 2016 | Phytoplankton | Camp water sample | Ferland J. / Grondin P.L. / Babin M. | Available |

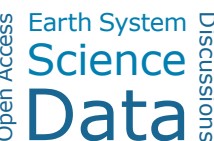

Table 2: Parameters measured during the Green Edge ice camp surveys. Parameters are ordered by alphabetical order and sampling year. *(continued)*

| Year | Parameter | Sampling method | Principal investigators | Processed |
|------|-----------|-----------------|-------------------------|-----------|
| 2016 | Phytoplankton (taxonomy) | Sediment Trap | Fortier L. / Lalande C. | Available |
| 2016 | Pigments | Camp water sample | Ras J. / Claustre H./Galindo V./ Rysgaard S. | Available |
| 2016 | Primary production | Camp water sample | Raimbault P. | Available |
| 2016 | Prokaryotic diversity | Camp water sample | Joux F. | Data not available yet |
| 2016 | Prokaryotic diversity | Sea ice core | Joux F. | Data not available yet |
| 2016 | Rrs $(0^+)$ | Optical radiometers profiling system | Becu G. / Babin M. | Available |
| 2016 | Salinity | Sea ice core | Galindo V. / Rysgaard S. | Available |
| 2016 | Scattering Coefficient | In-water IOP profiler | Becu G. / Babin M. | Available |
| 2016 | Sea ice concentration | Surface mode | Massicotte P. | Available |
| 2016 | Selenium | Collected organisms | Dufour F., Massé G., Ayotte P., Lemire M. | Available |
| 2016 | Silica Biogenic (BSi) | Camp water sample | Leynaert A./Moriceau B./ Leblanc K./Queguiner B. | Available |
| 2016 | Silica Biogenic (BSi) dissolution rate | Camp water sample | Moriceau B. | Available |
| 2016 | Silica Lithogenic (LSi) | Camp water sample | Leynaert A./Moriceau B./ Leblanc K./Queguiner B. | Data not available yet |
| 2016 | Silicate $Si(OH)_4$ - absorption kinetics | Camp water sample | Leynaert A. | Available |
| 2016 | Silica (uptake rate) | Camp water sample | Leynaert A. | Available |
| 2016 | $Si(OH)_4$ | Camp water sample | Leynaert A. / Moriceau B. | Available |
| 2016 | $Si(OH)_4$ | Camp water sample | Raimbault P. | Available |
| 2016 | $Si(OH)_4$ | Sea ice core | Raimbault P. | Available |
| 2016 | Snow depth | Camp snow sample | Galindo V./ Rysgaard S. | Available |
| 2016 | Spectral downwelling radiance angular distribution | Under-water sensor | Lambert-Girard S. / Leymarie E. | Available |
| 2016 | Spectral transmittance through ice | Surface mode | Verin G./Picard. G. | Available |
| 2016 | Surface spectral albedo | Surface mode | Verin G./Picard. G. | Available |
| 2016 | Suspended particulate material (SPM) | Camp water sample | Babin M. / Ferland J. | Available |
| 2016 | Swimmers | Sediment Trap | Fortier L. / Lalande C. | Available |
| 2016 | Synechococcus (abundance) | Camp water sample | Vaulot D. | Available |
| 2016 | Temperature | Sea ice core | Galindo V. / Rysgaard S. | Available |
| 2016 | Total organic carbon (TOC) | Camp water sample | Raimbault P. | Available |
| 2016 | Total organic carbon (TOC) and dissolved organic carbon (DOC) | Rosette | Panagiotopoulos C./ Sempéré R. | Available |
| 2016 | Total organic nitrogen (TON) | Camp water sample | Raimbault P. | Available |
| 2016 | Total organic phosphorus (TOP) | Camp water sample | Raimbault P. | Available |
| 2016 | Under-ice export fluxes of biogenic matter (fresh) | Sediment Trap | Fortier L. / Lalande C. | Available |
| 2016 | Upwelling Irradiance ($E_u(z)$) | Optical radiometers profiling system | Becu G. / Babin M. | Available |



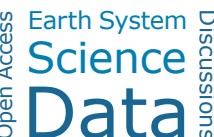

Table 2: Parameters measured during the Green Edge ice camp surveys. Parameters are ordered by alphabetical order and sampling year. *(continued)*

| Year | Parameter | Sampling method | Principal investigators | Processed |
|------|-----------|-----------------|-------------------------|-----------|
| 2016 | Upwelling radiance ($L_u(z)$) | Surface mode | Belanger S. / Goyens C. / Lambert-Girard S. | Data not available yet |
| 2016 | Upwelling radiance ($L_u(z)$) | Optical radiometers profiling system | Becu G. / Babin M. | Available |
| 2016 | Vertical profile of snow density | Surface mode | Verin G./Picard. G. | Available |
| 2016 | Vertical profile of Specific Surface Area | Surface mode | Verin G./Picard. G. | Available |
| 2016 | Virus (abundance) | Camp water sample | Joux F. | Available |
| 2016 | Wind Direction | Meteorological Tower | Massé G. | Available |
| 2016 | Wind Speed | Meteorological Tower | Massé G. | Available |
| 2016 | Zooplancton (Abundances) | Plankton net | Fortier L. / Aubry C | Available |
| 2016 | Zooplancton fecal pellet production rate | Plankton net | Fortier L. / Sampei M | Available |
| 2016 | Zooplancton grazing rate | Plankton net | Fortier L. / Sampei M | Available |
| 2016 | Zooplancton (Taxonomy) | Plankton Net | Fortier L. / Aubry C | Available |
| 2016 | Zooplankton vertical distribution | Underwater Vision Profiler (UVP) | Lagunas J. / Picheral M. | Available |



**Appendix A:  Surface tidal height**

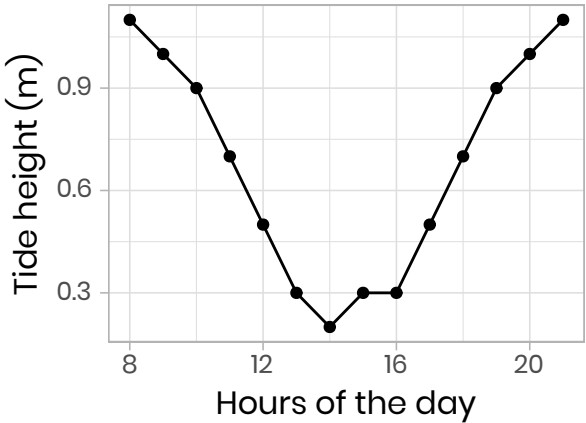

**Figure A1.** Surface tidal height versus time at Qikiqtarjuaq measured on 2015-06-09.

**Appendix B: GoPro Hero 4 photos**

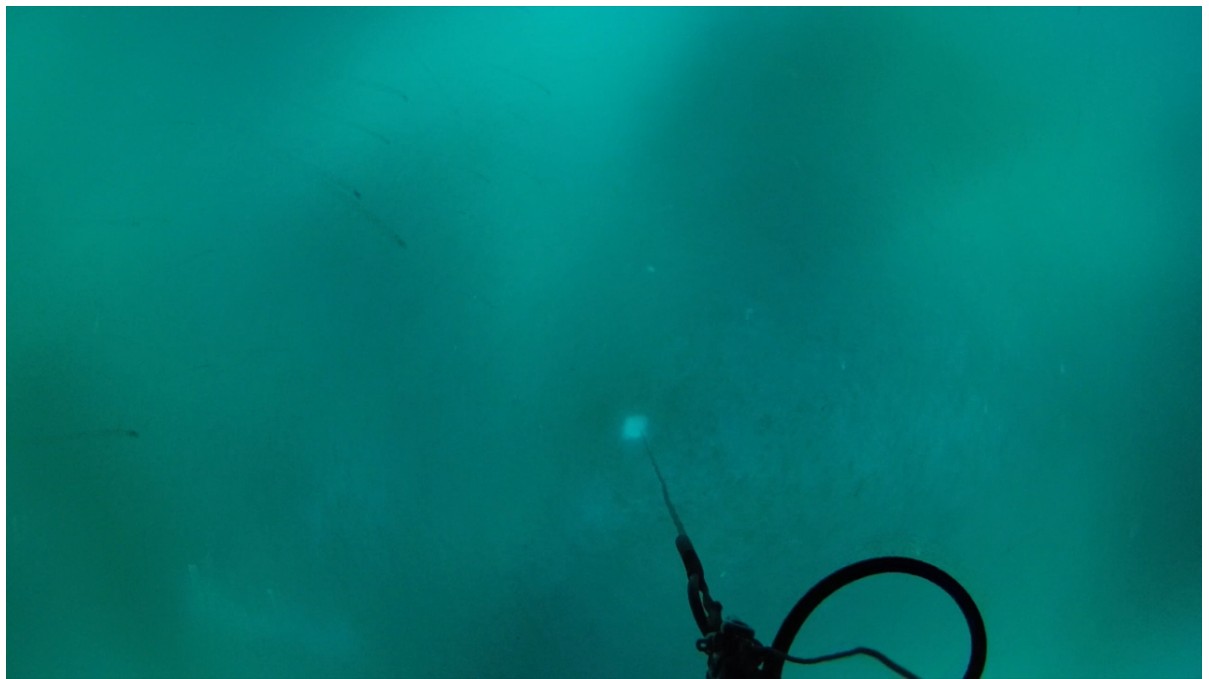

**Figure B1.** Video frame (00:58) from GoPro Hero 4 recording of C-OPS descent from 0 to 30 m, 18 May 2015 at the "low snow" hole. Note the streaks of nekton swimming across the upper left quadrant of the frame. Many planktons were seen in this profile, indicating an active under-ice community. A profile of the "high snow" hole on the same day, just 40 m away, showed no such plankton activity.

**Table B1.** Examples of GoPro Hero 4 photos at the low and high snow holes in 2015 demonstrating the spatial variability of the ice bottom across time and space.

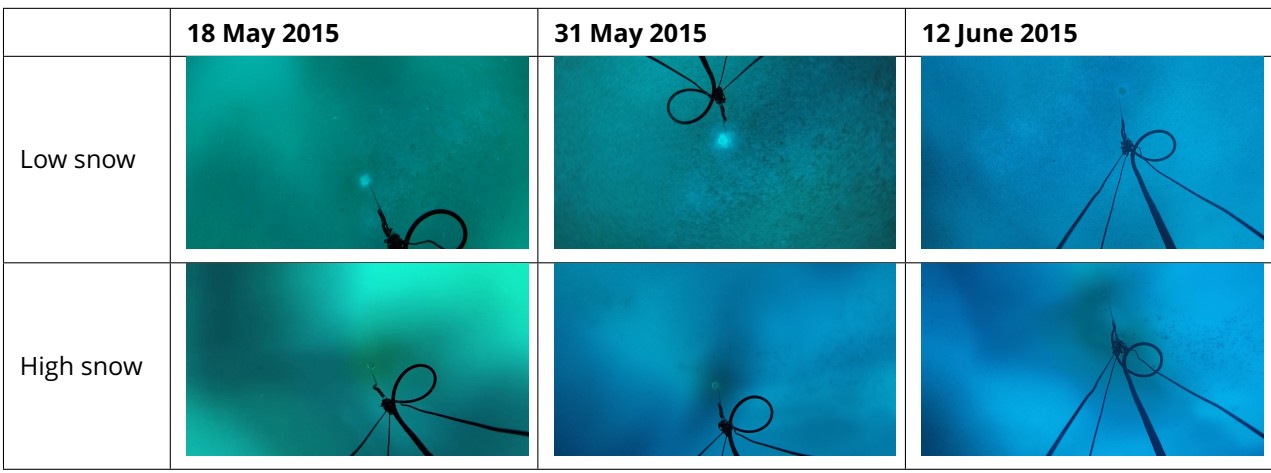

|  | **18 May 2015** | **31 May 2015** | **12 June 2015** |
|---|---|---|---|
| Low snow |  |  |  |
| High snow |  |  |  |



*Author contributions.* Ghislain Picard designed the snow optical measurements and participated in the 2015 campaign along with Gauthier Verin who sampled the 2015 and 2016 snow-related measurements. Anda Vladoiu, Caroline Sevigny and Dany Dumont deployed and Marie-Noëlle Houssais added her contribution to the analysis of the Self-Contained Autonomous MicroProfiler (SCAMP) on 23 June 2016 and quality-controlled, processed, analyzed and interpreted the data. Guislain Becu, Claudie Marec performed the setup and deployment of the CTD inside the tent in 2015. CTD setup and deployment was performed by José Lagunas, Christiane Dufresne, in 2016. Guislain Becu, Griet Neukermans, Eric Rehm, Simon Lambert-Girard and Laurent Oziel, Jade Larivière, Joannie Ferland, Julien Laliberté, performed the setup, calibration, and deployments of the ICE-Pro optical profiler outside the tent and the IOP frame inside the tent. Eric Rehm performed the 13-h tidal cycle measurements in 2015. Griet Neukermans and Eric Rehm deployed the GoPro Hero on the ICE-Pro. Claudie Marec performed the setup and installation of IFCB in the lab in 2015. Joannie Ferland performed the setup and installation of the IFCB in the lab in 2016. Joannie Ferland, Erin Reimer, Atsushi Matsuoka, Marie-Hélène Forget and Pierre-Luc Grondin performed the measurements. Pierre-Luc Grondin analyzed the data. Claudie Marec and José Lagunas performed the setup and deployment of an In-water profiler for particle size distribution and zooplankton vertical distribution (UVP Underwater Visio Profiler). Claudie Marec and José Lagunas performed setup and water sampling in both 2015 and 2016 campaigns. Claudie Marec was involved in the design and deployment of the ADCP in 2015, José Lagunas deployed the instrument in 2016. Atsushi Matsuoka coordinated the sampling strategy of discrete waters in terms of examining the linkages between optical and organic matter properties. Atsushi Matsuoka and Annick Bricaud wrote the protocols for both CDOM and particulate absorptions. For aCDOM, Atsushi Matsuoka, Joannie Ferland, Marie-Hélène Forget, Erin Reimer, and Pierre-Luc Grondin contributed to the measurements. For ap, Atsushi Matsuoka, Céline Dimier, Léo Lacour, Joséphine Ras, Mathieu Ardyna, Henry Bittig, Blanche St-Béat and Thomas Lacour contributed to the measurements. In 2015, particulate spectral absorption was also done by Lisa Matthes, Christine Quiring and Jens Ehn. Nicole Pogorzelec (who also did snow and ice salinity and overall chl-a filtrations in the field lab). Marie-Pier Amyot worked on tidying and uniformizing the data. Martí Galí ran the radiative transfer calculations and compared them to irradiance measurements taken on the ice camps. Lisa Matthes, Simon Lambert-Girard, Bob Hodgson, Jens Ehn, Nicole Pogorzelec and CJ Mundy designed and/or carried out the TriOS and ROV under-ice irradiance measurements Christos Panagiotopoulos and Richard Sempéré coordinated the sampling strategy for sugars/DOC and the analyses. Remi Amiraux collected the samples. Between October 2014 and July 2016, Éric Brossier and France Pinczon du Sel conducted measurements, collected clams, maintained equipment, kept a time-lapse photography record and represented the Greenedge team in Qikiqtarjuaq outside of the sampling season. Debra Christiansen Stowe coordinated logistics in Qikiqtarjuaq, in support of the 2016 ice camp. Makoto Sampei designed and curried copepods incubations to collect fecal pellets out at the ice camp in 2016. Makoto Sampei made microscopic observations on the collected fecal pellets in the laboratory. Sea ice and snow hemispherical directional reflectance were measured on the ice camp in 2015 by Sabine Marty and Clémence Goyens. The set-up was designed by Sabine Marty, Edouard Leymarie, Simon Bélanger and Clémence Goyens. They also processed and analyzed the data.

*Competing interests.* The authos declar no competing interests.



*Acknowledgements.* The GreenEdge project is funded by the following French and Canadian programs and agencies: ANR (Contract #111112), CNES (project #131425), IPEV (project #1164), CSA, Fondation Total, ArcticNet, LEFE and the French Arctic Initiative
(GreenEdge project). This project would not have been possible without the support of the Hamlet of Qikiqtarjuaq and the members of the community as well as the Inuksuit School and its Principal Jacqueline Arsenault. The project was conducted under the scientific coordination of the Canada Excellence Research Chair in Remote Sensing of Canada's new Arctic frontier and the CNRS & Université Laval Takuvik Joint International laboratory (UMI3376). The field campaign was successful thanks to the contribution of A. Wells, M. Benoît-Gagné, and E. Devred from the Takuvik laboratory as well as R. Hodgson from the University of
Manitoba. Pascale Bouruet-Aubertot and Yannis Cuypers who provided the SCAMP and contributed to the processing, quality control, analysis and interpretation of the data. We also thank Michel Gosselin, Québec-Océan, the CCGS Amundsen and the Polar Continental Shelf Program for their in-kind contribution to the logistic and scientific equipment. Thanks to Etienne Ouellet for IT support and data infrastructure management.



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
