# Peer review of "Green Edge ice camp campaigns: understanding the processes controlling the under-ice Arctic phytoplankton spring bloom"

_Earth System Science Data, 2019_

## Referee Comment (RC1) · Anonymous Referee #1 · 25 Oct 2019

The paper presents useful and complete data about two ice-camp sampling campaigns in the Arctic. The quality checks are appropriate and the process of reviewing the data is up-to-date and grants for usefulness of the data to other potential scientists. The data and methods are sufficiently described and well presented, excpet than for the productivity section (line 197-200), where the method and the protocol used are not well delineated. The presentation is of high quality and I don't see any inconsistencies that could raise suspects that the data are erroneous. As such, in my opinion, the data presented hold potential for being reused in the future for comparison and further elaboration.

---

## Referee Comment (RC2) · Anonymous Referee #2 · 30 Oct 2019

The authors produced an impressive, integrated data set. All the procedures are well explained and are supportive of their quality. I believe that any significant data set, in terms of representativeness and relevance of variables, complemented by clear description of procedures, is worth to be made accessible acknowledging the data generators for their willingness to share their data. This also when no intercalibration with other teams has been performed. I have only one question for the authors. Why they use the units of g kg-1 instead of PSU?

---

## Short Comment (SC1) · 2 Nov 2019

Thank you for the constructive comment. We will make sure that we precise the methods used for measuring production. Thank you for pointing it out.

Regards, Philippe
* * *

---

## Short Comment (SC2) · 2 Nov 2019

First, thank you for reviewing this data paper, it is much appreciated.

We have decided to use the "new" TEOS salinity definition. In the next paragraph, I am citing this website that defines it.

http://www.teos-10.org/

"A significant change compared with past practice is that TEOS-10 uses Absolute Salinity SA (mass fraction of salt in seawater) as opposed to Practical Salinity SP (which is essentially a measure of the conductivity of seawater) to describe the salt

content of seawater. Ocean salinities now have units of g/kg.

Absolute Salinity (g/kg) is an SI unit of concentration. The thermodynamic properties of seawater, such as density and enthalpy, are now correctly expressed as functions of Absolute Salinity rather than being functions of the conductivity of seawater. Spatial variations of the composition of seawater mean that Absolute Salinity is not simply proportional to Practical Salinity; TEOS-10 contains procedures to correct for these effects.

Importantly, while Absolute Salinity (g/kg) is the salinity variable that is needed in order to calculate density and other seawater properties, the salinity which should be archived in national data bases continues to be the measured salinity variable, Practical Salinity (PSS-78)."

With regards, Philippe

---

## Short Comment (SC3) · 2 Nov 2019

Hi again. Just to provide more information, the two types of salinity are available in the data. As we can see in the attached figure, they are very correlated to each other and the relationship follows the 1:1 line.

Regards, Philippe

[Figure]

**Fig. 1.**

---

## Author Comment (AC1) · 18 Nov 2019

Dear reviewer. As recommended, we have added the following paragraph to briefly give information on the methods used to measure primary production.

"Briefly, rates of carbon fixation (primary production), were measured using a dual 13C-15N isotopic technique (Raimbault1999). Water samples and ice melted was collected into three 600 ml polycarbonate bottles, previously rinsed with 10 % HCl, then with ultrapure Milli-Q water. Labelled 13C sodium bicarbonate ($NaH13CO3$ – 6 g, 250 mL-1 deionized water – 99 at % 13C, EURISOTOP) was added to each bottle in order to obtain $\approx$ 9.7% final enrichment (0.5 mL/580 mL-1 seawater). After addition of 13C-tracer

(H13CO3), samples were spiked with inorganic nitrogen labelled with 15N. Immediately after tracers addition, samples were fixed on an array placed under ice. Incubation was stopped after 24 hours and samples were immediately filtered on Whatman GF/F filters (25 mm diameter) pre combusted at 500°C. These filters were used to determine the final 15N/13C enrichment ratio in the particulate organic matter and the concentrations of particulate carbon and particulate nitrogen."

Please let us know if more information is required.

Regards, Philippe

---

## Author Response (AR1)

**Answers to the reviewers**

Dear Editor and reviewers, we first want to thank you for carefully evaluating the manuscript and giving us the opportunity to revise it accordingly. We carefully addressed each comment made by both reviewers. You will find below our point-by-point responses to each of these comments. Please find attached a clean and also a tracking changes versions of the manuscript. Note that for some unknown reason related to the editing tool we are using, the tracked version does not contain the changes made in the abstract. We hope that this version will be satisfactory and thank you for your time in this matter.

Yours sincerely,

Philippe Massicotte, Ph.D.

Research Associate

Takuvik Joint International Laboratory (UMI 3376)

Université Laval (Canada) & Centre National de la Recherche Scientifique (France)

Québec-Océan / Pavillon Alexandre-Vachon

1045, Ave de la Médecine, local 2064

Université Laval

Québec (Québec)

G1V 0A6 Canada

c.c. all the authors of the paper.

**Reviewer #1**

**Comment C1**

The paper presents useful and complete data about two ice-camp sampling campaigns in the Arctic. The quality checks are appropriate and the process of reviewing the data is up-to-date and grants for usefulness of the data to other potential scientists. The data and methods are sufficiently described and well presented, excpet than for the productivity section (line 197-200), where the method and the protocol used are not well delineated. The presentation is of high quality and I don't see any inconsistencies that could raise suspects that the data are erroneous. As such, in my opinion, the data presented hold potential for being reused in the future for comparison and further elaboration.

**Answer A1**

Thank you for the comments. We agree that the paper was missing information on the methods used to measure primary production. The following paragraph has been added to the paper (see the section entitled Phytoplankton).

*Briefly, rates of carbon fixation (primary production), were measured using a dual $^{13}C$-$^{15}N$ isotopic technique (Raimbault1999). Water samples and ice melted was collected into three 600 ml polycarbonate bottles, previously rinsed with 10 % HCl, then with ultrapure Milli-Q water. Labelled $^{13}C$ sodium bicarbonate (NaH$^{13}CO_3$ – 6 g, 250 mL-1 deionized water – 99 at % $^{13}C$, EURISOTOP) was added to each bottle in order to obtain ≈ 9.7% final enrichment (0.5 mL/580 mL-1 seawater). After the addition of $^{13}C$-tracer (H$^{13}CO_3$), samples were spiked with inorganic nitrogen labelled with $^{15}N$. Immediately after tracers addition, samples were fixed on an array placed under the ice. Incubation was stopped after 24 hours and samples were immediately filtered on Whatman GF/F filters (25 mm diameter) pre combusted at 500°C. These filters were used to determine the final $^{15}N/^{13}C$ enrichment ratio in the particulate organic matter and the concentrations of particulate carbon and particulate nitrogen.*

**Reviewer #2**

**Comment C2**

The authors produced an impressive, integrated data set. All the procedures are well explained and are supportive of their quality. I believe that any significant data set, in terms of representativeness and relevance of variables, complemented by clear description of procedures, is worth to be made accessible acknowledging the data generators for their willingness to share their data. This also when no intercalibration with other teams has been performed. I have only one question for the authors. Why they use the units of g kg-1 instead of PSU?

**Answer A2**

Thank you for the comments. We have decided to use the "new" TEOS salinity definition. We have specified everywhere in the text and updated the figure legend to replace *salinity* with *absolute salinity* ($S_A$) which is the new standard.

In the next paragraph, we are giving precisions on this new salinity standard.

http://www.teos-10.org/

"A significant change compared with past practice is that TEOS-10 uses Absolute

Salinity SA (mass fraction of salt in seawater) as opposed to Practical Salinity SP

(which is essentially a measure of the conductivity of seawater) to describe the salt content of seawater. Ocean salinities now have units of g/kg. Absolute Salinity (g/kg) is an SI unit of concentration. The thermodynamic properties of seawater, such as density and enthalpy, are now correctly expressed as functions of Absolute Salinity rather than being functions of the conductivity of seawater. Spatial variations of the composition of seawater mean that Absolute Salinity is not simply proportional to Practical Salinity; TEOS-10 contains procedures to correct for these effects.

Importantly, while Absolute Salinity (g/kg) is the salinity variable that is needed in order to calculate density and other seawater properties, the salinity which should be

archived in national data bases continues to be the measured salinity variable, Practical Salinity (PSS-78)."

Furthermore, the following graph shows that the PSU and absolute salinities are tightly correlated and that there are only very few differences between these two ways of measuring salinity.

[Figure]

[revised manuscript text omitted]
,\lambda)$, downwelling scalar irradiance, $E_{0d}(z,\lambda)$, and upwelling scalar irradiance, $E_{0u}(z,\lambda)$. These four hyperspectral radiometers (two planar RAMSES-ACC and two scalars RAMSES-ASC, TriOS GmbH, Germany) measured pressure and tilt internally and recorded irradiance spectra in the wavelength range from 320 to 950 nm at a resolution of 3.3 nm (190 channels). Transmitted irradiance was recorded along with vertical profiles by lowering the L-arm manually through a 20-inches auger hole with a

winch and 1.5-m aluminum poles extensions. In 2015, 17 vertical profiles were collected in 0.4 - 0.5-m depth steps from the ice bottom to a water depth of 18 m. In 2016, 11 profiles were recorded to a depth of 20 m under different sea ice surface conditions. Differences between planar and scalar PAR measurements were used to derive the downwelling average cosine, $\mu_d$, an index of the angular structure of the downwelling under-ice radiation field which, in practice, can be used to convert between downwelling scalar, $E_{0d}$, and planar, $E_d$, irradiance. The average cosine was smaller prior to snowmelt in 2015 compared to after snowmelt (~0.6 vs. 0.7), when melt ponds covered the ice surface in 2016 (Fig. 6). Further details about the sampling procedure, data processing and results can be found in Matthes2019.

*Inherent optical properties (IOP)*

IOPs measurements were made using an optical frame equipped with the physical and bio-optical sensors that were factory calibrated before each field campaign. A Seabird SBE-9 CTD measured temperature, absolute salinity, and pressure. A WetLabs AC-S was used for spectral beam attenuation ($c$, m$^{-1}$) and total absorption ($a$, m$^{-1}$) between 405 and 740 nm, and a BB9 (WetLabs) and a BB3 (WetLabs) were utilized for backscattering coefficients ($b_b$, m$^{-1}$) between 440 and 870 nm. During both campaigns, pure water calibration was performed for the AC-S sensor on each sampling day and linear regression of these calibration values as a function of time was computed for each wavelength of absorption and attenuation signals. Then, the offset applied during the data processing was taken on this linear regression at the exact date of the measurement. Figure 7 shows two vertical profiles of attenuation coefficients at different wavelengths acquired during pre-bloom and bloom conditions in 2016. One can see that during the bloom, attenuation

increased markedly in the 0-50 m surface layer due to higher phytoplankton biomass.

*Other optical measurements*

Other optical variables measured during both field campaigns included absorbance of particulate matter, absorbance of dissolved organic matter, snow and sea-ice transmittance, snow/ice hyperspectral and hyperangular hemispherical-directional-reflectance (Goyens2018) and surface spectral albedo (Verin et al., 2019, doi:10.5194/tc-2019-113Verin2019) (Table 2). Downwelling spectral irradiance above the surface (1°x1° 
[revised manuscript text omitted]